

# Improvements on the BRAMS wildfire-atmosphere modelling system

Isilda Cunha Menezes[1], Luiz Flávio Rodrigues[2], Karla M. Longo[2], Mateus Ferreira e Freitas[3], Saulo R. Freitas[2], Rodrigo Braz[2], Valter Ferreira de Oliveira[4], Sílvia Coelho[1], Ana Isabel Miranda[1]

[1]Center for Environmental and Marine Studies (CESAM), Department of Environment and Planning, University of Aveiro, Campus Universitário de Santiago, 3810-193 Aveiro, Portugal.
[2]Center for Weather Forecasting and Climate Studies (CPTEC), Department of Physics, National Institute for Space Research, Cachoeira Paulista, SP, Brazil.
[3]Multiuser Laboratory of High Performance Computing (LaMCAD), UFG Innovation Agency Building, Federal University of Goiás (UFG), Samambaia Campus, 74690-631, Goiânia, GO, Brazil.
[4]Finatec, Brasília-DF, Campus Darcy Ribeiro, Brazil.

*Correspondence to*: Isilda C. Menezes (isildacm@ua.pt), Saulo R. Freitas (saulo.r.de.freitas@gmail.com)

**Abstract.** Wildfire smoke significantly perturbs atmospheric composition and radiative balance, with implications for air quality, weather, and climate. Accurately simulating smoke–radiation–convection interactions remain a scientific challenge, particularly at meso-local scales. This study presents developments in the BRAMS v6.0 modelling system, including the integration of crown fire spread into SFIRE and dynamic coupling of fire-emitted smoke fluxes. These enhancements enable physically consistent simulations of wildfire behaviour, smoke emissions, and their radiative impacts.

The model couples fire spread and heat release to compute Fire Radiative Power (FRP), which drives smoke emissions in real time. These are fully integrated with aerosol–radiation interactions and atmospheric chemistry. The system was applied to the 15 October 2017 wildfire in central Portugal using high-resolution simulations.

Model performance was evaluated against MERRA-2 aerosol optical depth (AOD). Simulations reproduced key features of smoke transport and optical properties, including extinction and absorption coefficients at 400, 550, and 700 nm, as well as their spectral dependence. Results confirmed the dominance of organic carbon in extinction and validated the use of 550 nm as representative for smoke optical depth. Absorption reached $8\,\mathrm{m^{-1}}$ at 550 nm and led to vertical displacements of CAPE and CIN layers up to 200 m. Inversion layers responded to plume heating, exhibiting radiative lid effects that suppressed vertical mixing.

These findings demonstrate the potential of the enhanced BRAMS system to simulate coupled fire–atmosphere processes, contributing to improved forecasting of smoke behavior and understanding of wildfire-induced thermodynamic and radiative impacts.

**Graphical Abstracts.**





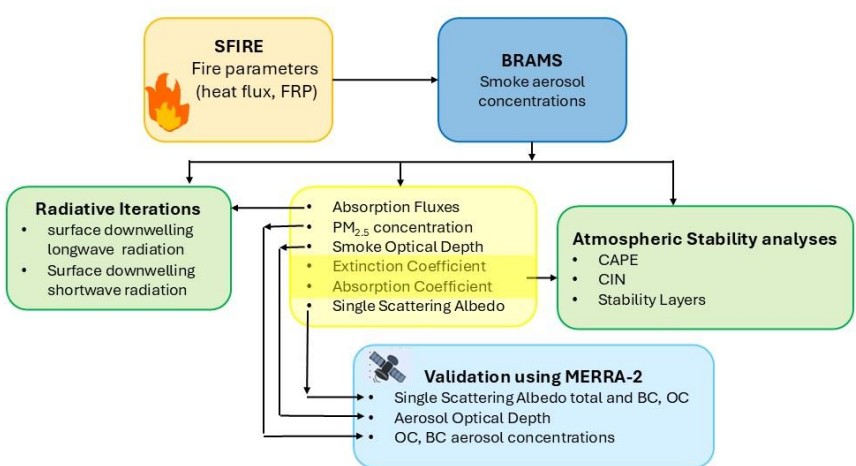

# 1 Introduction

Understanding wildfire emissions (Fernandes et al., 2022) is essential for accurately predicting air quality and radiative
impacts, particularly in complex terrains where smoke dispersion is strongly influenced by local meteorological dynamics.
Wildfires release large quantities of trace gases and aerosols that interact with atmospheric chemistry (Andreae et al., 2001;
Jaffe et al., 2008; Monks et al., 2012), modulate radiative transfer (Clinton et al., 2006; Menezes et al., 2024), reduce
visibility (Valente et al., 2007), and severely affect human health (Apte et al., 2018; Lopes et al., 2024; Miranda et al., 2010).
These impacts are especially difficult to quantify under the rapidly changing conditions typical of wildfire events, requiring
dynamic models capable of resolving fire–atmosphere feedback at high spatial and temporal resolutions.

Traditional approaches to estimating fire emissions in operational air quality models often rely on empirical methods, such as
pre-compiled inventories (e.g., GFED) or satellite-derived Fire Radiative Power (FRP), as used in the CAMS-GFAS
framework (CAMS, 2015). While computationally efficient, these methods do not simulate fire spread or associated heat
release, limiting their capacity to resolve the dynamic feedback that modulate local atmospheric stability and pollutant
dispersion. The Brazilian Biomass Burning Emission Model (3BEM) (Freitas et al., 2009; Longo et al., 2010) coupled to
BRAMS (Brazilian Regional Atmospheric Modeling System) follows this approach, estimating smoke emissions based on
FRP, surface type, and meteorology, but without coupling to fire dynamics. This approach is suitable for large-scale or long-
term assessments but lacks physical consistency under rapidly evolving fire conditions.

To address these limitations, BRAMS (Freitas et al., 2017; Menezes et al., 2021; Menezes, 2015) incorporates a physically
based fire–atmosphere interaction through the SFIRE model, which was recently enhanced with crown fire propagation and
dynamic smoke injection capabilities, as presented in this study. This integration enables simulations that explicitly resolve



surface and canopy fire spread, fuel consumption, and the associated release of heat and smoke over heterogeneous terrain. Fire Radiative Power (FRP) is not prescribed but internally computed from model-resolved sensible heat fluxes, allowing smoke emissions to evolve dynamically with fire intensity and meteorological conditions. The two-way coupling supports

feedback between combustion processes and local thermodynamics, which are critical for reproducing vertical smoke transport and smoke–radiation–convection interactions.

Compared to the WRF-SFIRE system (Allen et al., 2021; Kochanski et al., 2021; Mandel et al., 2014; Mandel et al., 2011), BRAMS shares a similar physical coupling architecture for simulating fire–atmosphere interactions. BRAMS originates from the RAMS model, a globally applicable mesoscale model initially developed in the United States. Over time, RAMS

was adapted with physical parameterizations better suited to represent processes relevant to tropical regions, leading to the development of BRAMS.

BRAMS has been extensively applied in South America, it remains a versatile model capable of simulating atmospheric processes across a wide range of latitudes, from tropical to mid-latitude conditions, in both hemispheres. The main differences between BRAMS and WRF lie in their physical parameterizations, vertical coordinate systems, and the way they

resolve the atmospheric base state. These structural differences influence how each model represents radiation, microphysics, turbulence, and surface–atmosphere interactions, particularly under conditions of strong vertical development such as those driven by wildfires. Notably, the inclusion of crown fire modeling in BRAMS offers enhanced realism in forested and mountainous regions where canopy combustion substantially increases heat release and aerosol injection heights.

The developments presented in this study were carried out within the FIRESMOKE project, a collaborative effort between the National Institute for Space Research (INPE), in Brazil, and the University of Aveiro, in Portugal. They include the integration of crown fire dynamics and the coupling of fire-emitted aerosols with the Coupled Chemistry Aerosol-Tracer Transport model (CCATT) (Longo et al., 2013), allowing for detailed simulation of smoke composition and transport. These developments are being incorporated into the forthcoming BRAMS v6.0.

Given the limited spatial and temporal resolution of satellite-based aerosol products—such as MODIS, VIIRS, and MERRA-2 reanalysis (MERRA-2, 2015)—and their gaps under cloudy or nighttime conditions, physically based models such as BRAMS provide a complementary approach for investigating wildfire smoke impacts. While BRAMS does not directly output aerosol optical properties, it provides detailed fields of smoke concentrations, aerosol speciation, and thermodynamic variables. These can be used to derive extinction, scattering, and absorption coefficients using Mie theory or other radiative

transfer frameworks, enabling the construction of spectrally and vertically resolved aerosol optical fields.

This capability is particularly relevant for evaluating the optical and radiative impacts of wildfire plumes, as illustrated in this study, which focuses on the Sertã wildfire that occurred on 15 October 2017 in a mountainous region of central Portugal (Menezes et al., 2024). Through high-resolution simulations with BRAMS, we quantify the vertical and spectral distribution of smoke-related aerosols and assess their influence on radiative fluxes and atmospheric stability. The complex terrain and



strong mountain–valley breezes in the study region significantly modulate smoke dispersion and vertical mixing, offering a challenging but highly relevant testbed for model validation.

By linking simulated fire behaviour and smoke emissions to column-integrated radiative effects, this work provides a physically consistent framework for analysing Smoke Optical Depth (SOD) and related extinction profiles. The approach also allows cross-comparison with satellite-derived Aerosol Optical Depth (AOD) products, improving the interpretation of

remote sensing data and advancing the representation of aerosol–radiation interactions in atmospheric models.

Section 2 describes the methodology and model developments, while Section 3 presents key results, including the performance of the model in simulating fire spread, smoke injection heights, PM2.5 concentrations, and radiative impacts. The findings highlight the value of two-way fire–atmosphere coupled systems for advancing our understanding of wildfire-induced air quality degradation and feedback on atmospheric dynamics.

## 95  2 Methodology

The following sections detail the advancements made in the BRAMS model, including the integration of a crown fire spread behaviour model and the implementation of FRP calculations within SFIRE. The processes for assimilating National Fuel Fire Laboratory (NFFL) fuel behaviour models and high-resolution terrain elevation data are also described, ensuring accurate nesting of the SFIRE grid within the BRAMS grid for two-way coupling of wildfire fluxes and emissions.

Furthermore, the methodology used to characterize particulate matter with an equivalent diameter less than 2.5 μm (PM2.5) released during the Sertã wildfire episode on 15 October 2017, as simulated by BRAMS, is thoroughly explained. This includes the validation and interaction of smoke with atmospheric radiation, as well as the specification of surface conditions required by SFIRE simulation.

### 2.1 BRAMS-SFIRE modelling system

The BRAMS is based on the Regional Atmospheric Modeling System (RAMS) developed at Colorado State University (CSU/USA). It is free software (CC-GPL), maintained by CPTEC/INPE, USP, and other institutions in Brazil and abroad. BRAMS and RAMS are versatile numerical weather prediction models, capable of simulating atmospheric circulations across a wide range of scales, from hemispheric to large eddy simulations (LES) of the planetary boundary layer. Significant improvements have been made to BRAMS to better represent key physical processes in tropical and subtropical regions,

alongside integrated atmospheric chemistry and aerosol processes. It also includes a state-of-the-art model for simulating the exchange of water, energy, momentum, and biogeochemical tracers between the atmosphere and the surface (Freitas et al., 2017). BRAMS solves the compressible, non-hydrostatic atmospheric equations described by Tripoli and Cotton (1982). It uses a one-way nesting scheme to perform downscaling on computational grids with increasing spatial resolution (Freitas et al., 2017).



Additionally, BRAMS incorporates a real-time assimilation method for the CCATT, focusing on the assimilation of trace gas and aerosol emissions from biomass burning, using FRP data derived from MODIS and GOES satellite products (Pereira et al., 2022; Pereira, 2013). Although the model typically underestimates emissions by up to 25%, it successfully captures around 90% of biomass burning activity, demonstrating high potential for near real-time monitoring of gases and aerosols emitted during biomass combustion. The FRP-based method improves emission estimations by 25% compared to traditional

methods, accurately capturing regional and local biomass burning patterns (Pereira, 2013).

BRAMS was coupled with the SFIRE model to simulate atmosphere and fire interactions (Mandel et al., 2011). The model was further enhanced by the authors through the inclusion of crown fire behaviour, as presented in this study. The communication between the BRAMS model and the SFIRE model is facilitated through an interface program. In this process, variables from BRAMS related to the basic state of the atmosphere are assimilated through this interface with

SFIRE. These variables include surface pressure, air moisture at 2 m, air temperature at 2 m, zonal and meridional wind components at the lowest vertical model levels, geopotential height, air density, microphysics, vegetation cover (including roughness and vegetation type), and heat fluxes.

When SFIRE is called for the first time step, the driver initializes SFIRE model by reading SFIRE's parameters from a name list, configuring flags, and setting the grid spacing. The high-resolution SFIRE domain is then integrated into the BRAMS

model domain by creating a grid centred on the ignition points, with a radius that encompasses the area predicted to be burned by the fire. Detailed input on fuel behaviour models and terrain height at high resolution must be provided, which are then read and interpolated onto the SFIRE grid domain. In subsequent steps, via the driver, SFIRE provides FRP to the BRAMS model after the ignition start time, contributing to the simulation of smoke-derived particulate matter concentration and its dispersion throughout the atmospheric column. Additionally, sensible and latent heat fluxes are integrated within the

BRAMS turbulence process, ensuring a realistic simulation of fire-smoke-induced energy transfer.

The BRAMS model offering two approaches for simulating smoke pollution, (1) the dispersion of smoke particulate matter derived from FRP, simulated by SFIRE and used as a passive tracer for atmospheric transport, and (2) a chemistry-coupled approach that integrates smoke particulate matter from SFIRE with emissions from the Biomass Burning Emission Model (3BEM), anthropogenic and biogenic inventories, and analysis data from the Copernicus Atmosphere Monitoring Service

(CAMS). This latter approach is implemented through the CCATT module of BRAMS, enabling more comprehensive simulations of atmospheric composition, pollutant transport, and concentration patterns. These dual methodologies enhance the model's capability to accurately represent the complex interactions between fire emissions and atmospheric chemistry.

## 2.2 New improvements on BRAMS-SFIRE

Several enhancements were made to this coupled BRAMS-SFIRE system, namely Figure 1 schematically shows these

improvements.





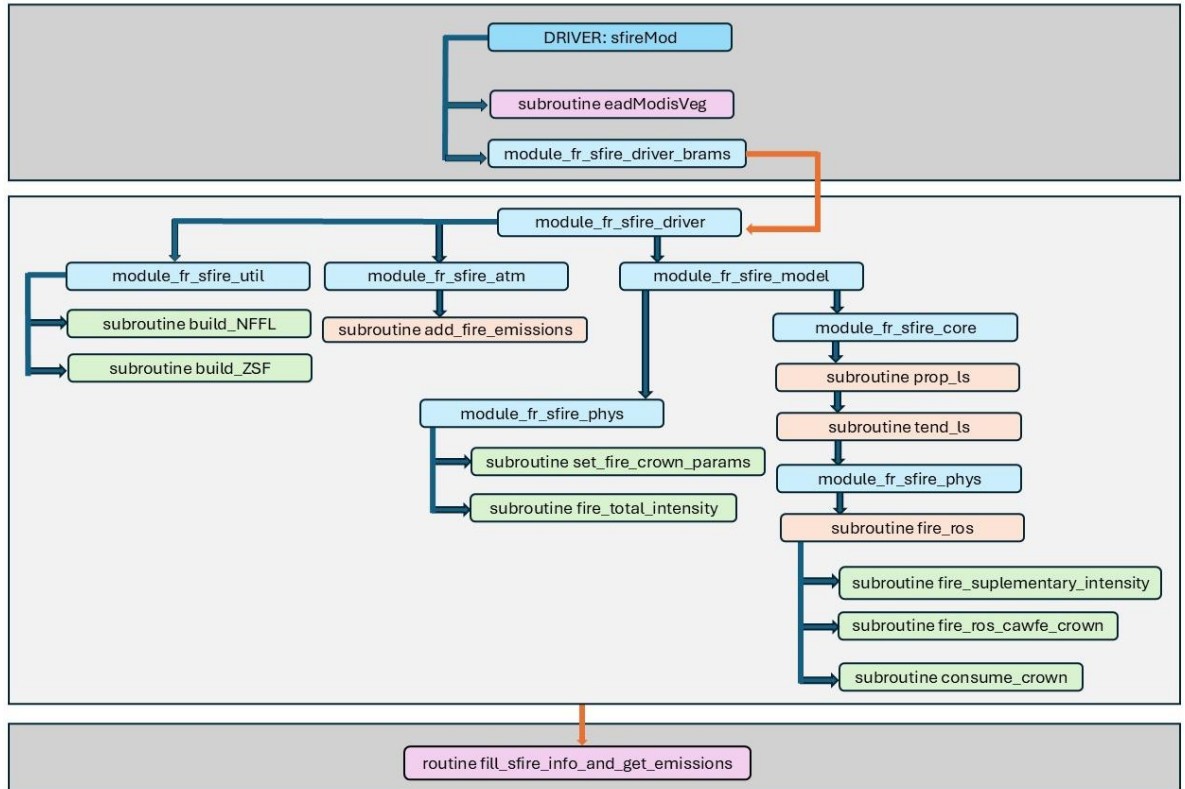

**Figure 1: Diagram illustrating the structure of the SFIRE model, with newly integrated subroutines denoted in light green boxes and called within the subroutines highlighted in light orange. The light blue boxes indicate the modules where these subroutines are implemented. The light pink boxes represent the subroutines responsible for integrating the BRAMS and SFIRE models in the context of the wildfire emissions.**

### 2.2.1 Implementation of the Crown Fire Behaviour Model

The SFIRE model (Mandel et al., 2009; Mandel et al., 2011), coupled with BRAMS (Freitas et al., 2017) by Menezes et al. (2021) and Menezes (2015), enables the analysis of surface fire behaviour, accounting for fuel bed type along with its thermal, mineral, and dendrometric characteristics, moisture content, topography, and atmospheric wind conditions. Significant improvements were made to this coupled model based on the work of Scott and Reinhardt (Scott and Reinhardt, 2001), by integrating a numerical model that predicts the spread and intensity of crown fires. This enhancement was incorporated into the SFIRE surface fire spread subroutines, as is illustrated in the scheme shown in Fig. 1. The new subroutines were developed to improve the prediction of fire spread and intensity in tree canopies and to perform the interpolation of NFFL fuel behaviour models and terrain elevation onto the SFIRE grid.





In the 'set_fire_crown_params' subroutine, the quasi-steady rate of spread is calculated for fuel behaviour model 10, which is a standardized vegetation fuel description. This model defines the physical and chemical properties of the fuel that are essential for computing fire spread rate, intensity, and overall fire behaviour. In the 'consume_crown' subroutine, the rate of active crow fire spread and the fraction of the canopy fuels consumed are computed. To determine these parameters, the subroutine calculates the equivalent rate of spread $R_{inicialization}$ (Eq. (1)), which depends on the critical (minimum) fireline intensity $I_{initialization}$ of the surface fire, and the heat release per unit area H. It is important to note that SFIRE is implemented using the United States customary system of units (e.g., BTU ft$^{-2}$, ft min$^{-1}$), and therefore, physical quantities derived or presented in this study follow that unit system unless otherwise specified.

$$R_{inicialization} = \frac{60 I_{initialization}}{H} \tag{1}$$

Where 60 is a conversion factor so that the units for I reduce to kW m$^{-1}$ (kJ m$^{-1}$ min$^{-1}$), and:

$$H = I_{r_{10}} t_r \quad \text{(in kJ m}^{-2}) \tag{2}$$

The symbol $I_{r_{10}}$ represents the reaction intensity for NFFL fuel behaviour model 10, while $t_r$ (Eq. (3)), represents the residence time for the same model (in minutes). The latter variable is a function of the surface-area-to-volume ratio (σ), as described below,

$$t_r = \frac{12.595}{\sigma_{10}} \tag{3}$$

The rate of active crown fire spread $R_{active}$ (Eq. (4)), set to 40 % of the observed 6.1-meter windspeed at midflame height, i.e., the average height within the flaming combustion zone where wind directly influences fire behavior. This height is commonly used in fire behavior modeling for standardization, particularly in the context of surface and crown fire interactions. Active crown fire spread, varies with wind speed, but is also influenced by slope and foliar moisture content (*FMC*, measured in percent), through the foliar moisture effect (*FME*) as defined by Van Wagner (Van Wagner, 1974, 1989, 1993),

$$R_{active} = 3.34 \left(\frac{FME}{FME_0}\right) (R_{10})_{40\%} \quad \text{(in m min}^{-1}) \tag{4}$$

with,



$$FME = \frac{(1.5 - 0.00275 FMC)^4}{460 + 25.9 FMC} \tag{5}$$


considering the ratio of *FME* to a reference value $FME_0 = 0.0007383$, established for a foliar moisture content (FMC) of 100 percent (Scott and Reinhardt, 2001).

Building upon the transition function concept proposed by Van Wagner (1989) that provides a scaling mechanism between surface and crown fire spread predictions, the model accounts for this process using Eq. (6).


$$R_{surface} \geq R_{inicialization} + 32.8084 \text{ (in ft min}^{-1}\text{)} \tag{6}$$

*CFB* (Eq. (7)) is a transition function known as the crown fraction burned, representing the fraction of available canopy fuels consumed during a fire. It is used to estimate the degree of crowning (Van Wagner, 1989) and to predict the resulting spread 205 rate in fire behaviour simulations. CFB values range from 0 for a surface fire to 1 for a fully active crown fire.

$$CFB = 1 - e^{-0.23(R_{surface} - R_{inicialization})} \tag{7}$$

The 'fire_ros' subroutine applies criteria to determine the initiation and sustained spread of crown fires, classifying a fire as 210 either surface fire or crown fire, without differentiating between passive and active crown fire. The approach is based on threshold criterion (Eq. (8)) involving the fireline intensity required to sustain crown fire behaviour,

$$I_{surface} > I_{initialization} \tag{8}$$

with $I_{surface}$ representing the Byram fireline intensity, and $I_{initialization}$ calculated within the fire_suplementary_intensity subroutine, as defined in Eq. (9).

$$I_{initialization} = \frac{CBH(460 + 25.9 FMC)^{3/2}}{100} \text{ (in kW m}^{-1}\text{)} \tag{9}$$

where *CBH* represents the canopy base height. $R_{10}$, refers to the surface fire spread rate for fuel behavior model 10, with its components calculated within the 'fire_ros_cawfe_crown' subroutine, as defined in Eq. (10).

$$R_{10} = R_{back_{10}} + R_{wind_{10}} + R_{slope_{10}} \text{ (in ft min}^{-1}\text{)} \tag{10}$$



By substituting Eq. (10) into Eq. (4) from the 'consume_crown' subroutine and incorporating Eq. (7) for crown fraction burned (*CFB*) from the same subroutine, Eq. (11) can be derived to calculate the total fire spread, considering Eq. (8).

$$R_{final} = \begin{cases} R_{surface} + CFB\left(R_{active} - R_{surface}\right) & with\ crown\ spread \\ R_{surface} & otherwise \end{cases} \quad \text{(in ft min}^{-1}\text{)} \tag{11}$$

The fire_total_intensity subroutine calculates the fire energy flux ($\phi$) originating from the combustion of dried fuel, as defined in Eq. (12),

$$\phi = \frac{W_{surface}}{dt} R_{final}\left(1 - \frac{FMC}{1-FMC}\right) H_{Low} \tag{12}$$

where $W_{surface}$ represents the initial total mass of surface fuel (kg m$^{-2}$), and $H_{Low}$ denotes the lower heat content of the fuel. This calculation, defined in Eq. (13), is performed to predict the fire intensity, $I_{final}$, accounting for the integration between the shrub/herbaceous layer and the tree canopy.

$$I_{final} = \frac{\left(H + (\phi(CFB))\right) R_{final}}{60} \quad \text{(in kW m}^{-1}\text{)} \tag{13}$$

**2.2.2 Implementation of FRP in SFIRE**

The following equations, implemented in module_fr_sfire_model, are used to estimate the radiative power emitted from a burning area $A$, based on the Stefan-Boltzmann law. The radiative power $P$ (Eq. (14)) is proportional to the emissivity $\epsilon$, the Stefan-Boltzmann constant $\sigma$, the burnt area $A$, and the fourth power of the temperature $T$ (K).

Alternatively, the equation can also be expressed in terms of the sensible heat flux $Qs$ (W m$^{-2}$), the mass of the burned 245    material $m$, and the specific heat capacity $c$ of dry wood, as shown in Eq. (15).

$$P = 10^{-6}(\epsilon . \sigma . A . T^4) \quad \text{(in MW)} \tag{14}$$

$$P = 10^{-6}\left(\epsilon . \sigma . A .\left(\frac{Q_s}{m.c}\right)^4\right) \quad \text{(in MW)} \tag{15}$$


The parameters used in this study are as follows: an emissivity value of 0.85 (Àgueda et al., 2010; Riggan et al., 2004), a specific heat capacity for dry wood estimated as $c = 1500$ J (kg K)$^{-1}$ (Radmanović et al., 2014), and the Stefan-Boltzmann constant as $\sigma = 5.67 . 10^{-8}$ . W (m$^2$ K$^4$)$^{-1}$.





### 2.2.3 Fuel Models and Terrain Data Assimilation Subroutine Implementation

The build_NFFL and build_ZSF subroutines are responsible for the integration of NFFL fuel behaviour models and high-resolution terrain elevation data into the SFIRE computational domain. This process is carried out following the procedure detailed below:

- Data Input and Configuration: The subroutines read user-provided input files, generated in accordance with the methodology described by Menezes et al. (2021) and Menezes (2015). One file contains categorical information on NFFL fuel models, while the other provides terrain elevation data. During this step, key parameters such as grid resolution, initial coordinates, geospatial constants, and the dimensions of both datasets are defined.

- Coordinate Calculation: Geodetic coordinates (latitude and longitude) corresponding to the input data grids are calculated based on the predefined cell size.

- Data Interpolation: The input fuel and terrain data are interpolated onto the SFIRE grid by identifying the nearest neighbouring grid nodes. This interpolation employs the Haversine formula to calculate geodesic distances, ensuring that the curvature of the Earth is properly accounted for in the spatial mapping.

- Result Storage: The resulting interpolated datasets are saved in binary format for efficient access and subsequent use during the simulation workflow.

### 2.2.4 Coupling SFIRE Outputs with the BRAMS Framework for Wildfire Emissions Simulation

The fill_sfire_info_and_get_emissions subroutine integrate the FRP generated by the SFIRE domain simulation into the coarse-resolution grid of the BRAMS model, enabling the calculation of wildfire emissions. It performs the following key tasks:

- Iterates over the SFIRE grid to identify cells with positive FRP, burned area and burn duration. These cells are classified as valid active fire points.

- Stores fire data: For each valid fire point, the corresponding information is stored in an array, indexed by geographic coordinates.

- Calls the subroutine "get_emission_in_global_brams_grid", which:
  - Maps each fire point from the SFIRE grid onto the global BRAMS grid.
  - Associates each fire point with a vegetation type, based on land cover (COS) data provided within the BRAMS grid.
  - Calculates burned biomass and emissions of various particulate matter species (e.g., PM10, PM2.5), using emission factors and the flaming fraction, which distinguishes between flaming and smoldering phases.

- Aggregates emissions: The calculated emissions are summed into the respective BRAMS grid cells.

- Performs statistical calculations: Computes emission statistics, such as the mean and standard deviation of FRP and burned area for each BRAMS grid cell.



This workflow ensures a seamless integration between fire dynamics modelled by SFIRE and the atmospheric process represented in BRAMS, enabling accurate simulation of wildfire emissions and their atmospheric impacts. In this study, only BRAMS passive tracer dispersion capabilities are used, focusing on the transport and dispersion of tracers without engaging the full atmospheric chemistry module. This approach allows for detailed analysis of pollutant transport patterns while minimizing computational costs.

Rosário et al. (2025) developed seasonal Aerosol Spectral Optical Properties (ASOP) models for aerosol mixtures affecting the atmosphere over the Iberian Peninsula. These models were implemented within the BRAMS framework for improved optical properties of particulate material emitted by a wildfire simulation using the SFIRE model coupled to BRAMS. Based on a long-term (2003–2023) characterization of aerosol microphysical properties, optical parameters were derived using Mie theory applied to AERONET sun photometer data. The ASOP models represent typical regional aerosol scenarios, including smoke, dust, and background aerosol mixtures. Spatial representativeness was evaluated through monthly correlations between MERRA-2 and AERONET data, enabling the identification of areas predominantly influenced by each model. This approach enhances aerosol–radiative transfer simulations in BRAMS over the Iberian Peninsula, improving predictions of thermodynamic profiles and surface temperatures, while reducing uncertainties in the representation of aerosol optical properties.

### 2.3 Data requirements

### 2.3.1 Surface data used as input in SFIRE

To simulate the spread of surface and crown fires using SFIRE, high-resolution NFFL fuel behaviour models (Anderson, 1982) and terrain elevation data on a regular grid are required. In this study, NFFL fuel model data were derived from the Portuguese National Forest Inventory 6 (Menezes et al., 2021). Additional inputs are also needed for the "namelist.fire" file, including physical, thermal, chemical, and mineral specifications of surface vegetation (Menezes et al., 2021; Menezes, 2015), as well as of tree canopy fuel within the simulation domain (such as canopy base height (CBH) and crown foliar moisture content (FMC)) for 13 NFFL fuel models.

CBH and FMC values were assigned using GIS methods by intersecting primary and secondary land data with five dominant forestry species (Table 1) and cross-referenced with the 13 NFFL fuel models. These data were part of 7,964 field plots of the 2006 National Forest Inventory, provided by the Portuguese Institute for Nature Conservation and Forests (ICNF, 2015), and adopted from the work of Menezes (Menezes, 2015) based on measurements conducted in October 2010 in the Évora district—part of the Alentejo region in southern Portugal, characterized by agro-silvopastoral landscapes known as Montado.

Although FMC values from international literature were considered, they were ultimately excluded from this study, as they may not accurately represent the climatic, edaphic, and physiological conditions of Portuguese ecosystems. The FMC values presented in Table 1 were therefore used to do this classification.



**Table 1: Average crown foliar moisture content (in %) for the 13 NFFL fuel models based on field data from five dominant forestry species (Alentejo, Portugal; October 2010)** (Menezes, 2015)**.**

| Tree species | FMC |
|---|---|
| *Pinus pinaster* Aiton | 126.89 |
| *Pinus pinea* L. | 129.62 |
| *Eucalipto globulus* Labill | 118.51 |
| *Quercus rotundifolia* | 63.37 |
| *Quercus suber* L. | 72.32 |

Following the same methodology, CBH was determined. CBH is influenced by factors such as forest management practices, stand density, tree age, and species composition. In Portugal, CBH values can vary significantly. The specific average ranges, calculated using only the same five tree species, for each of the 13 NFFL fuel models, are detailed in Table 2.

**Table 2: Average canopy base height (in meters) for the 13 NFFL fuel models based on field data from five dominant forestry species (Alentejo, Portugal; October 2010)** (Menezes, 2015)**.**

| Fuel Models | CBH |
|---|---|
| NFFL 1 | 6.43 |
| NFFL 2 | 8.28 |
| NFFL 3 | 8.11 |
| NFFL 4 | 8.02 |
| NFFL 5 | 10.04 |
| NFFL 6 | 8.94 |
| NFFL 7 | 9.67 |
| NFFL 8 | 7.36 |
| NFFL 9 | 7.32 |
| NFFL 10 | 7.60 |
| NFFL 11 | 7.02 |
| NFFL 12 | 3.02 |
| NFFL 13 | 0.62 |



### 2.3.2 Surface and meteorological data used as input in BRAMS

To establish the initial atmospheric state, BRAMS incorporated several datasets, including land cover type, soil type, the Normalized Difference Vegetation Index (NDVI), weekly sea surface temperatures, daily soil moisture, and soil temperature. The NDVI was derived from 15-day MODIS composite images spanning 2001 to 2002 (Moreira et al., 2013), while weekly sea surface temperature data were obtained from datasets distributed by (Reynolds et al., 2002). Daily soil moisture, an operational product of CPTEC/INPE based on rainfall estimated from TRMM (Gevaerd and Freitas, 2006), was also integrated. The initialization of soil temperature was based on air temperature values from the first level of the BRAMS atmospheric model.

Terrain elevation data with a spatial resolution of 30 arc seconds (approximately 1 km) in latitude and longitude were sourced from the United States Geological Survey's Earth Resources Observation Systems data centre (Gesch et al., 1999) and assimilated during the model initialization.

The atmospheric fields, including zonal and meridional wind, air temperature, geopotential height, and relative humidity, used for both the initialization and boundary conditions, were obtained from the ECMWF Reanalysis v5 (ERA5, 2019). These data were available at 37 vertical pressure levels and 6-hour intervals on a regular grid and were interpolated to the BRAMS grid to ensure consistency between the initial and boundary conditions.

### 2.3.3 Data used for model validation

Aerosol concentration, extinction coefficients, and AOD values were retrieved from the Modern-Era Retrospective analysis for Research and Applications, version 2 (MERRA-2, 2015) dataset. This dataset assimilates observations from a variety of instruments, including satellite-based radiometers, to provide global, long-term atmospheric reanalysis data.

MERRA-2 offers a spatial resolution of $0.5° \times 0.625°$ and includes 72 hybrid-eta vertical levels extending from the surface to 0.01 hPa (Gelaro et al., 2017). Its temporal resolution is 3 hours, consistent across the various atmospheric and aerosol variables. The MERRA-2 collections used in this study were obtained from the Global Modeling and Assimilation Office (GMAO) (MERRA-2, 2015).

### 2.4 Comprehensive Study of SOD and Atmospheric Properties

### 2.4.1 Spectral SOD analysis and validation

In this study, the characterization of the SOD generated during the Sertã wildfire focused on analysing its magnitude, spectral dependence, and interaction with atmospheric radiation. To this end, the influence of meteorological conditions on SOD was assessed, considering variables such as air temperature, air humidity and wind speed.

SOD was estimated following Seinfeld and Pandis (Seinfeld and Pandis, 1998), assuming smoke particles to be monodisperse and spherical. PM2.5 concentrations ($\mu g\ m^{-3}$), simulated by BRAMS to represent wildfire smoke, were converted to particle number concentrations using factors established in the literature. These factors relate mass



concentration to optical depth by accounting for the density and refractive index of wildfire smoke particles. Since SOD is defined as the vertical integral of aerosol extinction (i.e., the sum of absorption and scattering), the contribution from each atmospheric layer was calculated using Eq. (16).

$$SOD = \int_0^\infty Q_{ext}(r_e, \lambda) \cdot n(r_e, z) \cdot \sigma(r_e) dz \tag{16}$$

where:

- $Q_{ext}(r_e, \lambda)$: Extinction efficiency (obtained from Mie theory, based on the particle size and refractive index);
- $n(r_e, z)$: Particle number concentration as a function of effective radius $(r_e)$ and height $(z)$;
- $\sigma(r_e)$: Particle cross-sectional area;
- $dz$: Atmospheric layer thickness.

For each vertical layer, the PM2.5 mass concentration is converted into particle number concentration (Eq. (17)) using an
375 assumed aerosol density of 1.35 g cm⁻³ (Reid et al., 2005), which is representative of typical wildfire smoke composition (Rosário et al., 2013), and a predefined particle size (effective radius) of 0.065 μm (Reid et al., 1998; Reid et al., 2005) consistent with wildfire aerosol characteristics.

This fixed aerosol density and particle radius are essential for converting PM2.5 mass concentrations (e.g., in kg m⁻³ or mg m⁻³) into particle number concentrations (particles m⁻³), which are required to calculate extinction, scattering, and absorption
coefficients through Mie theory.

$$n = \frac{C_m}{\rho V_p} \tag{17}$$

*where:*

- $n$: particle number concentration (particles m⁻³);
- $C_m$ : mass concentration (kg m⁻³);
- $\rho$: aerosol density (kg m⁻³);
- $V_p$: volume of a particle (m³), calculated as shown in Eq. (18):

$$V_p = \frac{4}{3}\pi r^3 \tag{18}$$

with $r$ being the effective radius of the particle (in meters).





The specific extinction efficiency is computed using Mie theory as a function of particle radius, wavelength, and complex refractive index. A wavelength-dependent refractive index was applied, consistent with physical trends reported in the literature (Anderson, 1982; Reid et al., 2005). The real part decreased slightly, and the imaginary part increased with wavelength, reflecting stronger absorption at longer wavelengths. The refractive indices used were: $1.55 + 0.02i$ at 400 nm, $1.50 + 0.03i$ at 550 nm, and $1.47 + 0.04i$ at 700 nm.

The outputs include the extinction efficiency ($Q_{ext}$), scattering efficiency ($Q_{sca}$), absorption efficiency ($Q_{abs}$), the Single Scattering Albedo (SSA), and the Asymmetry Parameter (g).

This study analysed three representative wavelengths from the visible spectrum: 400 nm (blue), which is more sensitive to smaller particles and where organic carbon (OC) absorption is significant; 550 nm (green), a commonly used reference in atmospheric studies; and 700 nm (red), where black carbon (BC) absorption is dominant, though overall absorption is lower than at shorter wavelengths.

SOD was calculated for each wavelength by vertically integrating extinction and absorption coefficients along both meridional and zonal cross-sections. The resulting SOD values were analysed in relation to thermodynamic parameters. Additionally, the influence of spectrally resolved smoke absorption on radiative transfer was examined, with emphasis on the visible to near-infrared range, to better understand the wavelength-dependent radiative impacts of smoke aerosols.

To ensure consistency with observational data, simulated SOD at 550 nm was directly compared with MERRA-2 AOD at the same wavelength. This spectral band is widely adopted as a standard reference in aerosol monitoring and ensures compatibility between model output and satellite-derived products.

Simulated SSA was compared with MERRA-2 SSA for BC and OC (Aerosol Optical Thickness (AOT) at 500 nm), as well as with the total aerosol SSA from MERRA-2. Furthermore, SOD at 550 nm was evaluated against MERRA-2 BC and OC extinction AOT. Simulated PM2.5 concentrations were also compared with MERRA-2 BC and OC mass concentrations.

### 2.4.2 Absorption of Solar Radiation by Smoke and Radiative Budget Impacts

The radiative balance was analysed by comparing simulations results with and without fire, specifically considering surface downwelling shortwave radiation (W m$^{-2}$), and surface downwelling longwave radiation (W m$^{-2}$). These accumulated energy fluxes are used to evaluate the radiation budget under smoke conditions at the surface. The smoke-absorbed shortwave radiation at 550 nm (F(550 nm)$_{abs}$, W m$^{-2}$) was estimated based on the heating potential of the fire plume and the vertical profile of aerosol absorption using the following relation expressed in Eq. (19):

$$F(550nm)_{abs} = H_{sen}\left(\int_0^\infty Q_{abs}(r_e,\lambda) \cdot n(r_e,z) \cdot \sigma(r_e)dz\right) \tag{19}$$

Where $H_{sen}$ is the surface sensible heat flux associated with the fire.



### 2.4.3 Vertical Profiles of Aerosol Optical Properties and Atmospheric Stability

Atmospheric stability was assessed specifically within smoke-affected layers of the atmospheric column. The analysis focuses on detecting thermal inversions by examining vertical profiles of potential temperature and relative humidity in regions influenced by simulated smoke, under both fire and no-fire scenarios. A thermal inversion is identified when both potential temperature and relative humidity increase with height.

To evaluate the impact of smoke on atmospheric stability, these thermodynamic profiles were analysed in conjunction with

vertical distributions of smoke extinction and absorption coefficients at 550 nm. This approach highlights the interaction between radiation and fine particulate matter (PM2.5), and the resulting effects on local temperature structure.

Furthermore, Convective Available Potential Energy (CAPE) and Convective Inhibition (CIN) were compared for smoke and non-smoke conditions. Differences in these parameters (ΔCAPE and ΔCIN) were computed and represented as vertical profiles, providing insights into how smoke alters convective dynamics and overall atmospheric stability.

### 3 Results and discussion

In the following sections, the burned area resulting from the wildfire episode in Sertã is compared between simulations using the BRAMS and WRF models, in order to assess fire spread performance under forested and complex topographic conditions. Simulated SOD is evaluated against MERRA-2 AOD to assess spatial and temporal coherence. Column-integrated extinction coefficients at 550 nm are also compared with extinction derived from MERRA-2 BC and OC aerosol

components, in order to verify whether the simulated optical processes reflect the dominant aerosol types observed.

The simulated SSA is compared with MERRA-2 SSA, along with simulated and MERRA-2 aerosol mass concentrations, to validate whether the simulated aerosol composition—particularly the relative abundance of BC and OC—is consistent with observed radiative behaviour. This comparison also aims to verify whether the simulated mass profiles of BC, OC, and PM2.5 correspond to the resulting optical properties. Frequency distributions of particulate matter concentrations across the

simulation domain are analysed to assess the spatial variability and intensity of aerosol loading. Additionally, the impact of smoke absorption on surface downwelling shortwave and longwave radiation is evaluated.

The optical effects on atmospheric stability layers are investigated through the analysis of CAPE and CIN. Vertical profiles of extinction and absorption coefficients are further examined in relation to the development of thermal inversion layers, in order to determine whether the model accurately reproduces the optical signature of smoke in radiative transfer simulations.

### 3.1 BRAMS and WRF model differences in burned area representation and their implications

Significant differences in the simulated burned area for the same Sertã wildfire event, as depicted in Fig. 2, were observed between BRAMS-SFIRE and the WRF-SFIRE results previously reported by Menezes et al. (Menezes, I. C., Lopes, D., Fernandes, A. P., Borrego, C., Viegas, D. X., Miranda, 2024). BRAMS-SFIRE simulated a broader westward spread, while



WRF-SFIRE produced a more confined footprint, extending further north. These discrepancies arise from differences in
model configuration, fire propagation schemes, and spatial resolution.

First, although WRF-SFIRE resolves fire spread at a finer horizontal resolution (20 m) than BRAMS-SFIRE (200 m),
BRAMS exhibits stronger sensitivity to terrain-induced effects due to its topographic assimilation and interaction with the
atmospheric fields at coarser grid scales (2 km in diagnostics). This can amplify upslope spread, especially on west-facing
slopes, where heating of unburned fuels by the flame front is intensified. Additionally, BRAMS employed a default rate of
spread approximately ten times greater than that used in WRF, resulting in more aggressive lateral fire expansion,
particularly in regions where wind-driven constraints are weaker.

Second, resolution differences in output diagnostics further influence the apparent burned area. WRF's higher resolution
(20m) output preserves more spatial detail, limiting smoothing of the burned footprint. Conversely, the BRAMS burned area
(200 m resolution) was accumulated and plotted in BRAMS 2 km grid resolution, which introduces spatial averaging and
may artificially enlarge the apparent extent of fire-affected regions.

Moreover, the delayed onset of burned area in BRAMS-SFIRE when using lower propagation rates is not solely due to
ignition misalignment, but rather a result of the interaction between the fire spread rate and the coarser fire grid resolution
(200 m). Unlike WRF-SFIRE, which operates at a much finer 20 m resolution for fire propagation, BRAMS requires the
flame front to traverse larger grid cells to ignite neighbouring areas. This means that if the default rate of spread is reduced
without compensating for grid size, the fire may fail to reach adjacent cells in a timely manner, leading to an
underrepresentation of burned area in the early stages. Therefore, a balanced calibration is needed between the fire spread
rate and grid resolution in BRAMS to ensure that the ignition front progresses consistently with observed timelines. While
finer resolution in WRF (20 m) inherently provides more accurate spatial detail in fire development (Menezes et al., 2024),
BRAMS relies on a more approximate representation, making it sensitive to how rate-of-spread parameters are tuned relative
to its mesh scale.

In summary, while WRF-SFIRE offers finer-scale detail in fire perimeter shape, BRAMS-SFIRE exhibits stronger coupling
between fire dynamics, topography, and atmospheric feedbacks. For prognostic simulations of fire spread and smoke
transport in complex terrain, BRAMS-SFIRE may provide enhanced physical realism — provided its rate of spread is
properly calibrated to avoid over propagation, particularly in orographically enhanced directions.





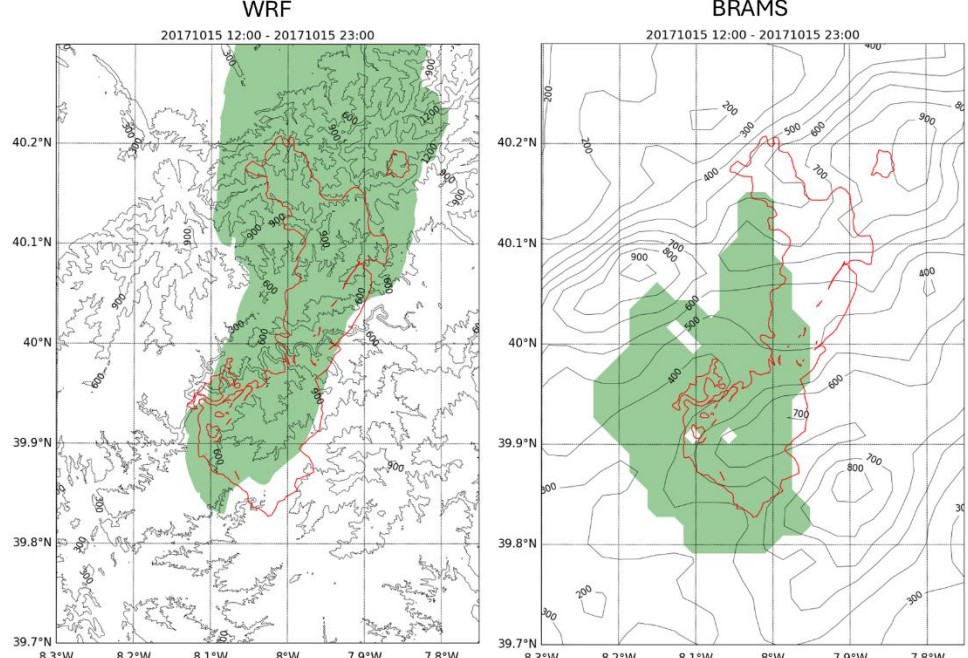

**Figure 2: Comparison of simulated burned area over high-resolution topography from the (a) WRF-SFIRE and (b) BRAMS-SFIRE models for the wildfire event on 15 October 2017, accumulated between 12:00 and 23:00 UTC. Green-shaded regions represent the cumulative burned area, while red contours delineate the observed fire perimeter.**

## 3.2 Optical properties comparison between SOD and AOD and TOA from MERRA-2

Forest fires, during their propagation, consume biomass fuel models and release high concentrations of particles and trace gases into the atmosphere. Primary aerosols are directly emitted from the flames, while secondary aerosols form in the atmosphere through chemical reactions involving precursor gases. However, primary particles can serve as condensation nuclei or "seeds" for secondary particle formation through aging processes, resulting in internally mixed aerosols with complex and evolving chemical compositions as they are transported within the smoke plume.

Primary particles include soot, predominantly composed of elemental carbon (BC), along with ash and mineral components derived from the burned biomass. These materials may exist as distinct particles or aggregate into larger mixed particles during combustion. Depending on their composition and surface properties, these aerosols can be hydrophilic (attracting water, enhancing growth in humid conditions) or hydrophobic (repelling water), affecting their interaction with radiation and their ability to act as cloud condensation nuclei (CCN).

As these particles are transported within the smoke plume, they undergo aging processes that alter their physical and chemical properties. These include: (1) Coagulation — collisions and merging with other particles, leading to increased size





and complexity; (2) Surface reactions – interactions with atmospheric gases such as $SO_2$, $NO_x$, and volatile organic compounds (VOCs), including terpenes and isoprenes (emitted by vegetation during natural biological processes) and

aromatic compounds (from biomass combustion). These reactions lead to the formation of sulfates, nitrates, and organic coatings; (3) Oxidation — particularly of BC, resulting in the acquisition of oxygen-containing functional groups that modify its optical properties and enhance hydrophilicity.

In parallel, VOCs are oxidized in the atmosphere to produce oxygenated organic compounds, such as carboxylic acids (e.g., formic acid, acetic acid) and aldehydes (e.g., formaldehyde), which contribute to the formation and growth of secondary

organic aerosols (SOA). These compounds can condense onto primary particles, forming coatings that significantly modify their optical and hygroscopic properties.

Moreover, photochemical reactions involving VOCs and oxidants such as ozone ($O_3$) and hydroxyl radicals (OH) also lead to secondary particle formation. Not all secondary aerosols are of local fire origin; VOCs and precursor gases transported from urban pollution sources or natural vegetation in other regions may also contribute further to increasing the complexity

of the smoke plume and its optical properties.

Both primary and secondary aerosols interact with solar radiation through scattering and absorption of shortwave (visible and near-infrared) radiation, reducing direct sunlight at the surface while enhancing diffuse radiation (Yamasoe et al., 2006). These particles also absorb and re-emit longwave (thermal infrared) radiation. While BC strongly contributes to absorption, sulfates and organic aerosols primarily enhance scattering, collectively influencing the atmospheric radiative energy balance

(Yamasoe et al., 2006). The net result often includes local atmospheric heating, particularly in layers rich in BC.

These radiative interactions are governed by Mie theory, as most smoke particle sizes fall within the Mie regime (approximately 0.1 μm to 1 μm), where the particle diameter is comparable to the wavelength of incident radiation. Smaller particles (<< 0.1 μm) scatter more efficiently at shorter wavelengths (e.g., blue light) and are better described by Rayleigh theory, which assumes particle sizes much smaller than the radiation wavelength. In contrast, larger particles exhibit non-

selective scattering and remain within the Mie regime, with their radiative behaviour less dependent on wavelength

Understanding the spectral behaviour of smoke aerosols is essential for interpreting optical depth measurements. In this context, both the simulated spectrally integrated SOD and the MERRA-2 AOD represent the attenuation of solar radiation due to the presence of aerosols in the atmosphere. However, there are conceptual differences that affect their comparison. SOD corresponds specifically to the fraction of AOD attributed to smoke particles generated by wildfires and biomass

burning, while MERRA-2 AOD includes all aerosol types present in the atmosphere, such as mineral dust, sulfates, nitrates, sea salt, and soot.

When SOD ≤ AOD, it indicates that smoke aerosols constitute only a portion of the total aerosol load. In contrast, when SOD ≈ AOD, it suggests that the majority of aerosols in the region and time analysed originate from biomass burning, which typically occurs during intense fire events with limited interference from other aerosol sources. Conversely, when SOD ≪

AOD, it implies a significant contribution from non-smoke aerosol sources, such as mineral dust (e.g., Saharan intrusions), industrial pollution (sulfates), or marine aerosols.





Fig. 3 shows the spatial patterns of SOD and AOD at 15:00, 18:00 and 21:00 UTC on 15 October 2017. This figure shows that between this period both SOD and AOD increased substantially, reflecting the intensification and vertical development of the smoke plume associated with large-scale forest fires. The early afternoon conditions show a more confined plume with

moderate optical depth, whereas in the evening, the plume becomes broader and optically thicker, indicating stronger fire activity, enhanced vertical injection, and increased aerosol loading.

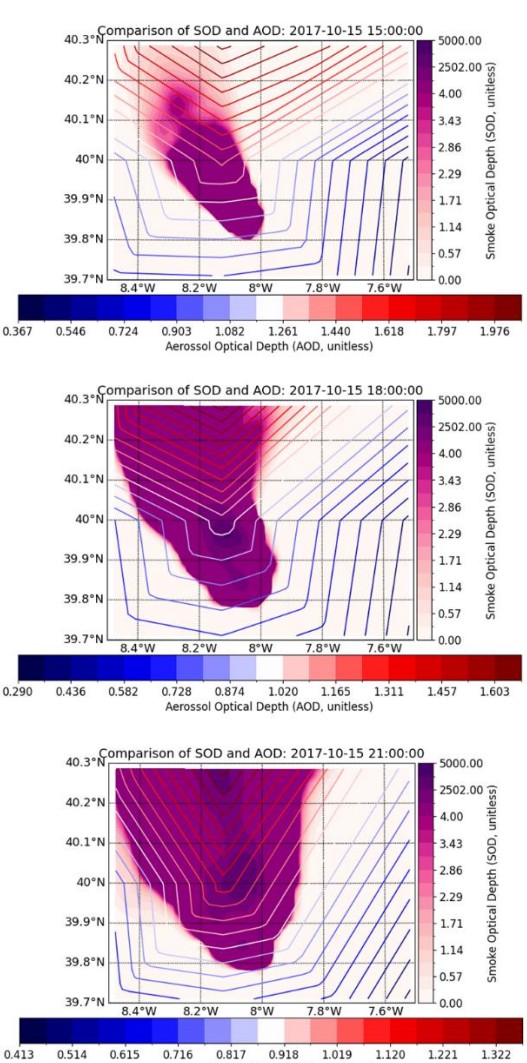

**Figure 3: Spatial distribution of simulated spectrally integrated SOD (colour shaded) over the study region at 15:00 UTC,**

**18:00UTC and 21:00 UTC on 15 October 2017, during the intense Sertã wildfire episode, with MERRA-2 AOD (colour contour) superimposed.**



MERRA-2 is an assimilated product that applies spatial and temporal smoothing. Its AOD reflects spatial averages and incorporates aerosols from multiple sources. As such, it does not respond instantaneously to abrupt emission events like intense wildfires with strong vertical injections. This limitation can result in an underestimation of peak aerosol optical depth in regions of active burning.

In contrast, the simulated SOD captures the freshly emitted smoke plume directly, with high local concentrations and vertical injections into elevated atmospheric layers, leading to high column-integrated values. These peaks may not be fully captured by MERRA-2. However, consistency is observed in peripheral regions without active fire, where no direct smoke emissions are present — the simulated SOD in these areas closely matches MERRA-2 AOD. This agreement supports the validity of the SOD calculation methodology, indicating that the model is not systematically overestimating aerosol load but is realistically representing fire intensity and smoke distribution.

An additional factor contributing to the discrepancy between simulated SOD and MERRA-2 AOD is the spatial resolution of the datasets. While the simulations were performed at a horizontal resolution of 2 km, MERRA-2 AOD has a native resolution of approximately 50 km. As a result, sharp gradients and localized peaks in aerosol loading associated with intense fire activity are significantly smoothed in the MERRA-2 product. This spatial averaging leads to an attenuation of AOD values in regions of active burning, whereas the high-resolution simulation retains the fine-scale structure and intensity of the smoke plume.

It is also important to recognize that the monodisperse approach adopted in this study — where all particles are assumed to be spherical, with identical fixed effective radius and density — inherently promotes higher SOD values in certain situations. This is particularly evident in the fire front, where $PM_{2.5}$ concentrations are extremely high and the total mass is directly converted into a large number of small particles. Because smaller particles have lower individual volumes, more particles per unit mass are generated, increasing the number concentration and, consequently, the extinction coefficient. This effect enhances the calculated SOD in these regions.

Furthermore, it is important to distinguish between the thermal radiation emitted by the fire itself and the attenuation of incoming solar radiation quantified by AOD and SOD. Fires emit thermal infrared radiation due to highly exothermic oxidation reactions involving biomass and oxygen. This thermal radiation is generated locally and contributes to atmospheric heating in the combustion zone. In contrast, AOD and SOD quantify the extinction of solar (shortwave) radiation caused by aerosol scattering and absorption and are therefore associated with changes in the atmospheric radiative balance induced by smoke particles — not with the direct release of energy from combustion.

The thermal radiation emitted by wildfires occurs primarily in the longwave (infrared) spectrum and can be expressed as a radiative flux (W m⁻²). Although this longwave emission does not directly interact with incoming solar radiation, it significantly alters the local thermodynamic environment — enhancing plume buoyancy, promoting aerosol dispersion, and facilitating the formation of secondary aerosols. These processes indirectly influence the transmission of solar radiation by modifying the spatial and vertical distribution of scattering and absorbing aerosols. Thus, thermal radiation from fire plays a





critical role in the radiative energy budget, both through its direct longwave component and through its indirect effects on shortwave radiative transfer via aerosol–radiation interactions.

Fig. 4 shows the spatial distributions of BC, OC, and PM2.5 concentrations at 15:00 and 21:00 UTC on 15 October 2017. Consistent with the nature of biomass burning emissions, OC concentrations were found to be substantially higher than those

of BC, indicating dominance of smoldering combustion processes during the event. This combustion phase is typically associated with lower temperatures and the release of large amounts of volatile organic compounds, which condense to form organic aerosols. As expected, OC concentrations exceeded BC by a factor of 10 or more, in line with previous observational studies.

The model outputs also showed significantly higher peak concentrations than those observed in the coarser MERRA-2

reanalysis (~50 km resolution), which tends to smooth out local emission gradients. The enhanced detail in BRAMS-SFIRE results from its finer horizontal resolution (2 km), which captures subgrid-scale variability critical for localized emission events. Furthermore, the spatial coherence between OC/BC and PM2.5 fields in both datasets confirms that carbonaceous aerosols are the dominant constituents of the aerosol mass within the smoke plume, with OC contributing the most to the PM2.5 burden.






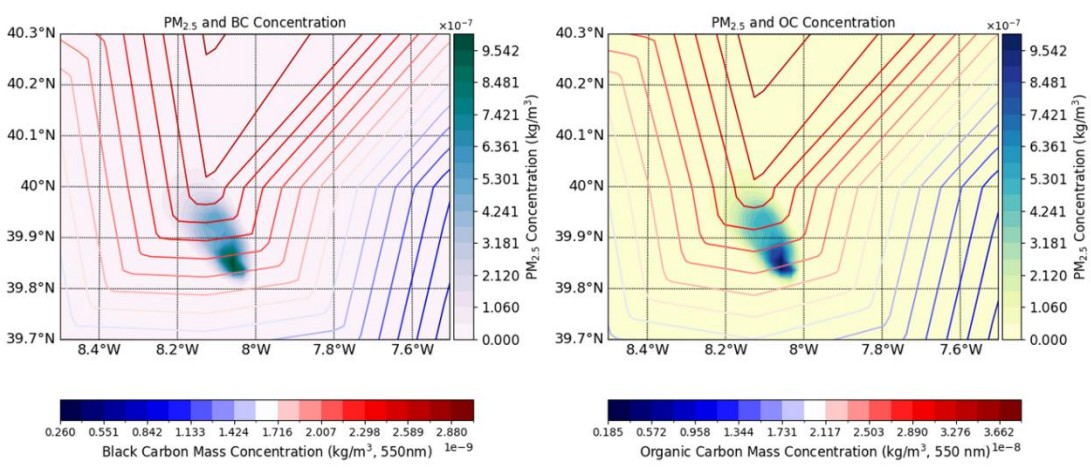

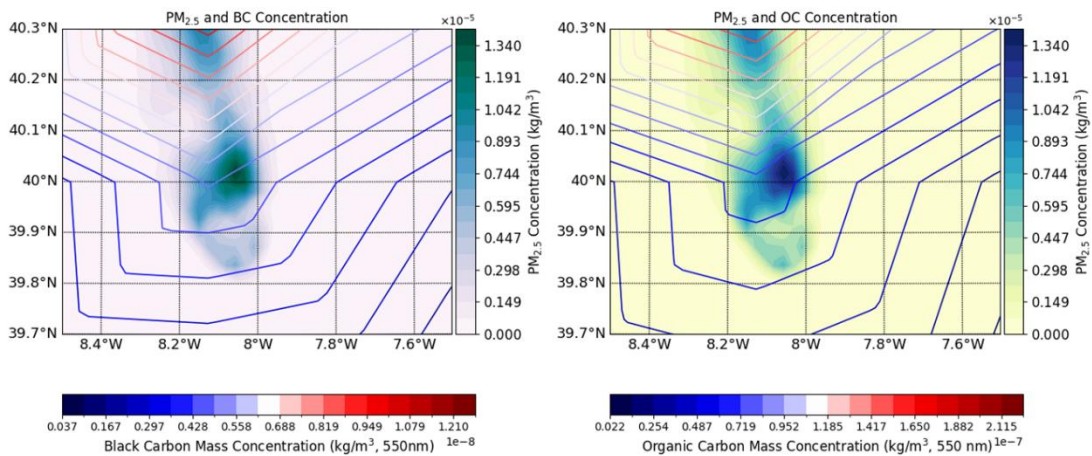

**Figure 4: Spatial distribution of PM2.5concentrations (kg·m⁻³, contours) overlaid on BC (left panel) and OC (right panel) mass concentrations (kg·m⁻³, colour shading) at 15:00 and 18:00 UTC on 15 October 2017.**


Fig. 5 further explores the optical impacts of these aerosols by comparing the MERRA-2 AOT for BC and OC against the simulated SOD at 550 nm wavelength. This wavelength is widely used in satellite remote sensing due to its sensitivity to fine-mode aerosol extinction, including both scattering and absorbing components. The results indicate that OC is the dominant contributor to the total extinction at this wavelength, reinforcing its role in shaping the smoke plume's radiative



behaviour. The 550 nm SOD from BRAMS-SFIRE closely matches the structure of spectrally integrated SOD, confirming
its suitability as a proxy for visible-band optical impacts of smoke.

**Figure 5: Comparison between simulated SOD (contours) and MERRA-2 extinction AOT at 550 nm for BC (left panel) and OC**
**(right panel), at 18:00 UTC on 15 October 2017.**

To evaluate the radiative implications of these aerosol fields, we compared the modelled SSA with that from MERRA-2 at
550 nm in Fig. 6. SSA is a key variable in aerosol radiative forcing studies, representing the relative importance of scattering
versus absorption. Lower SSA values imply stronger absorption (typically due to BC-rich emissions), while higher values
indicate more scattering-dominated aerosols such as sulphates or aged organics.

The comparisons are presented at three different times (15:00, 18:00 and 21:00 UTC), with the left panels contrasting model
SSA with MERRA-2 estimates derived from BC+OC AOT, and the right panels using MERRA-2 SSA from total AOT.
Across all times, BRAMS-SFIRE resolves spatially sharp gradients in SSA, with values as low as 0.38–0.50 in the plume
core, consistent with dense, freshly emitted smoke from crown fire phases. MERRA-2 SSA, particularly when derived from
BC+OC, displays a smoother structure and systematically higher values (typically >0.5), a likely result of lower spatial
resolution and climatological mixing assumptions in the reanalysis.

When comparing with MERRA-2 SSA based on total AOT (right panels), even larger discrepancies arise. These are due to
the inclusion of additional aerosol types (e.g., sulphates, dust) that increase the overall scattering component. The model's



ability to capture low SSA regions coinciding with fire hot spots and peak AOD zones underscores its internal consistency
and detailed physical coupling between emissions, transport, and radiative processes.

Taken together, Figures 4, 5, and 6 highlight the strength of high-resolution fire-atmosphere models like BRAMS-SFIRE in capturing the chemical composition and radiative behaviour of smoke plumes. Nonetheless, systematic biases in the SSA magnitude — potentially arising from uncertainties in aerosol mixing states, refractive index assumptions, or aging processes — call for further validation using observational datasets with high temporal and spatial resolution.




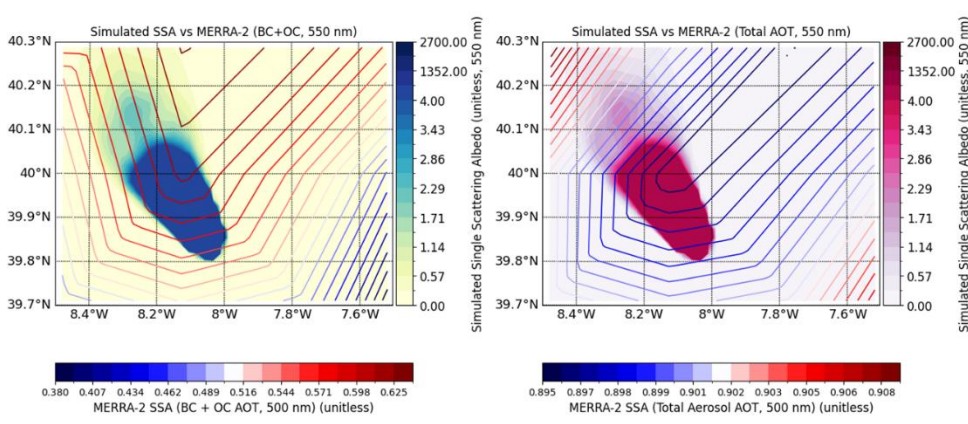

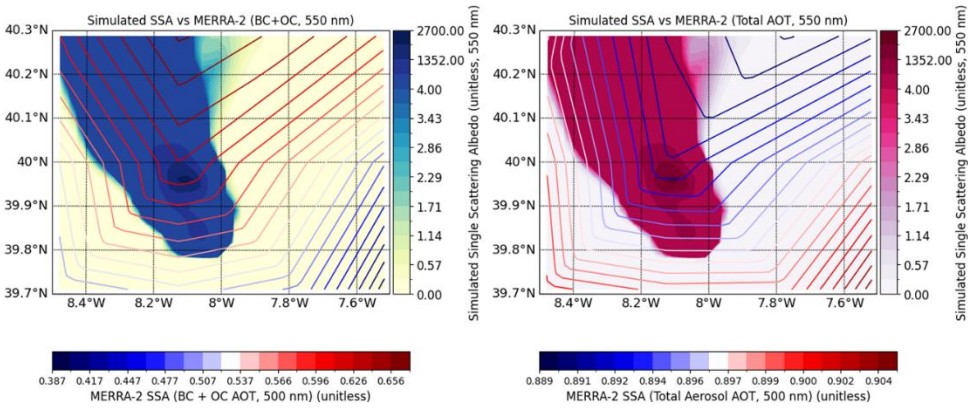

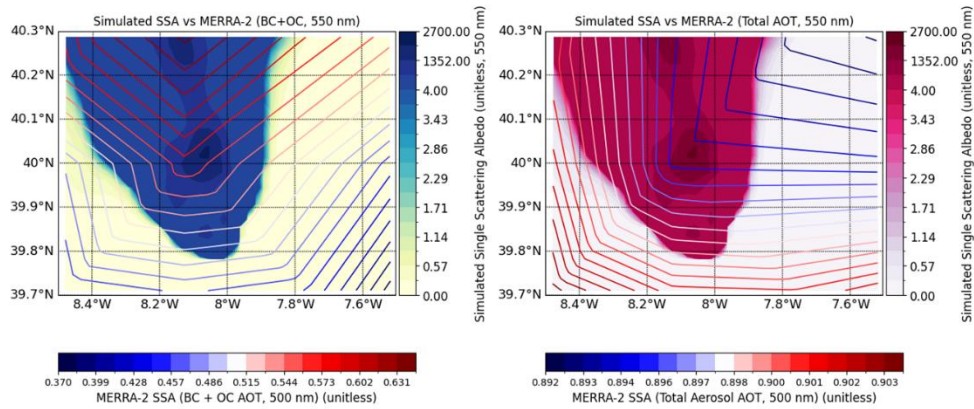



**Figure 6: Spatial distribution of simulated SSA at 550 nm from BRAMS-SFIRE, overlaid with SSA contours derived from MERRA-2 reanalysis for the wildfire plume on 15 October 2017, at 15:00, 18:00, and 21:00 UTC. Left panels show MERRA-2 SSA calculated from BC and OC AOT, while right panels represent SSA derived from total AOT (all species).**

### 3.3 Temporal Evolution and Distribution of PM2.5 Concentrations simulated

Fig. 7 shows histograms of the frequency distribution of PM2.5 concentrations (µg m⁻³) across the simulation domain on 15 October 2017. The frequency on the vertical axis represents the number of grid points registering concentrations within each bin. The use of a logarithmic scale on the Y-axis enhances the visibility of extreme values, even when they occur infrequently.

Across all time steps, the PM2.5 distribution is markedly right-skewed, with the majority of grid cells exhibiting relatively low concentrations (e.g., below 1,000 µg m⁻³), while a long tail extends toward extremely high values exceeding $10^7$ µg m⁻³. This indicates a highly inhomogeneous plume structure, dominated by localized hotspots of intense emissions.

At 13:00 UTC, most concentrations remain below 10,000 µg m⁻³, with only a few high-end outliers. As the fire evolves, the distributions widen — particularly at 15:00, 16:00, and 19:00 UTC—suggesting that more grid cells are being affected by elevated PM2.5 levels as the plume intensifies and expands. By 21:00–22:00 UTC, the tail persists but shows a modest decline in the frequency of extreme values, possibly indicating partial dispersion or reduced emissions due to diurnal evolution or fuel exhaustion.

The most extreme PM2.5 values, often exceeding $10^7$ µg m⁻³, likely originate from cells directly above the fire front, where emissions are injected into small vertical volumes. While such peaks are physically plausible, they may be artificially enhanced by model limitations such as coarse vertical resolution or insufficient representation of turbulent mixing.







**Figure 7: Frequency distribution of PM2.5 concentrations (µg/m³) across the simulation domain on 15 October 2017 at different hours (13:00, 14:00, 19:00 and 22:00 UTC). Each histogram shows the number of grid cells registering PM$_{2.5}$ values within each concentration bin. The vertical axis is plotted on a logarithmic scale to highlight rare but extreme values.**

### 3.4 Atmospheric Stability and Thermodynamic Modulation by Aerosol–Radiation Interactions in Wildfire Events

Atmospheric stability is primarily governed by the vertical temperature gradient, or lapse rate. The behaviour of an air parcel depends on how rapidly it cools adiabatically compared to the ambient temperature profile. If a rising parcel cools more slowly than the environment, it becomes less dense and buoyant, accelerating upward. Conversely, if it becomes denser than the surrounding air, it will descend. When vertical displacements of air parcels are amplified by buoyancy forces, the atmosphere is considered unstable, favouring convection, cloud formation, and potentially severe weather. In contrast, if the parcel is cooler and denser than its environment, vertical motion is suppressed, indicating a stable atmosphere.



Stability is evaluated by comparing the environmental lapse rate with the theoretical adiabatic lapse rates: the Dry Adiabatic Lapse Rate (DALR ≈ 9.8 K km$^{-1}$) and the Moist Adiabatic Lapse Rate (MALR, typically 4–7 K km$^{-1}$, depending on humidity). Strongly stable conditions are found during temperature inversions, where temperature increases with altitude. These often occur near the surface at night or under large-scale subsidence, resulting in a subsidence inversion typically between 1–3 km altitude. Such inversions act as barriers to vertical mixing and are crucial for air pollution accumulation.

When moist air parcels rise, they initially follow the DALR until reaching the Lifting Condensation Level (LCL), where condensation begins. The release of latent heat during condensation (an exothermic process) partially offsets adiabatic cooling, increasing the parcel's buoyancy and shifting it to follow the MALR. If sufficient energy is available, the parcel reaches the Level of Free Convection (LFC) and may continue to ascend to the Equilibrium Level (EL), near the tropopause.

The atmospheric layer closest to the surface is the Planetary Boundary Layer (PBL), where turbulence and mixing are most pronounced due to surface heating and friction. The structure and height of the PBL vary diurnally and are strongly influenced by radiative forcing and atmospheric stability. During the daytime, especially in wildfire regions, intense surface heating and smoke-induced turbulence may erode stable layers, promoting deep mixing.

These vertical motions have significant implications for aerosol dispersion. In stable conditions, especially under inversions, particulate matter accumulates near the surface due to suppressed mixing. Under unstable, convective conditions, particles are efficiently mixed and vertically transported.

CAPE quantifies the potential energy that a rising air parcel can convert into kinetic energy if lifted adiabatically. Higher CAPE values suggest stronger updrafts and a greater likelihood of convective development, enhancing vertical mixing and pollutant dispersion. CIN, on the other hand, represents the energy barrier that must be overcome for a parcel to initiate free convection. Large CIN values suppress vertical motion and contribute to the accumulation of pollutants and aerosols near the surface.

The regions impacted by these two energy terms are bounded by two key levels: the LFC and the EL. Below the LFC, the parcel is cooler than its environment, requiring energy input to rise—this energy defines the CIN. Above the LFC, the parcel is positively buoyant, contributing to CAPE. These estimates often assume pseudo-adiabatic processes (no entrainment and removal of condensed water), allowing latent heat release to maintain saturation.

In wildfire scenarios, evaluating CAPE and CIN become particularly relevant for assessing pyroconvection. Wildfires inject substantial heat and moisture into the atmosphere, potentially reducing local CIN (by warming the PBL) and allowing parcels to reach their LFC. When background CAPE is significant, this can trigger the formation of convective clouds (flammagenitus or pyrocumulus), and in extreme cases, deep convective events such as pyroCb (pyrocumulonimbus). Studies have shown that intense fire heat fluxes increase CAPE and reduce CIN, facilitating the vertical transport of smoke and the development of fire-induced storms.

In addition, turbulent fluxes generated by rising air parcels and plume dynamics influence pollutant dispersion. Parcels moving upward often carry higher aerosol concentrations, resulting in well-defined plume cores. This distribution can often




be approximated by Gaussian profiles in horizontal cross-sections, in agreement with Fick's law of diffusion, where the diffusion flux is proportional to the gradient in species concentration.

The interaction of aerosol particles with radiation is governed by two fundamental optical processes: scattering and absorption. Scattering occurs when incident electromagnetic radiation is redirected by particles. In the case of Mie scattering,
relevant when the particle diameter is comparable to the wavelength of the incident radiation, light penetrates the particle, and the internal refractive index gradients cause the redirection of light. OC aerosols, being efficient scatterers, predominantly affect shortwave (SW) radiation in the visible spectrum.

Absorption involves the uptake of photon energy by electrons in the atoms or molecules of the aerosol particle. The absorbed energy excites electrons to higher energy levels, and as they relax to their ground states, the energy is primarily dissipated
through non-radiative processes, increasing the internal kinetic energy of the particle. This manifests macroscopically as thermal heating of the particle, i.e., the conversion of radiant energy into thermal energy. BC is a strong absorber in the visible and near-infrared spectra and significantly contributes to atmospheric heating.

While both OC and BC interact with radiation in the UV–visible range (0.3–0.7 μm), the direct radiative effect of aerosols on longwave (LW) radiation is generally weaker than that on shortwave (SW) radiation.
The absorption of SW radiation by BC-rich aerosol layers elevates local temperatures, promoting thermal stability. This may lead to or enhance thermal inversions, thereby increasing CIN and inhibiting convection and cloud formation. In contrast, OC scattering reduces incoming SW radiation, cools the surface and reducing the vertical temperature gradient. This cooling may decrease CAPE, reducing the buoyant energy available for upward motion.

This thermodynamic modulation by aerosol–radiation interactions is further illustrated in Fig. 8, which presents the
differences in shortwave and longwave surface radiative fluxes between simulations with and without fire at 16:00, 17:00, and 21:00 UTC on 15 October 2017. In these panels, particularly in Figure (a), smoke is shown to cause a localized increase in longwave radiation (green-yellow to light-coloured areas), which is consistent with thermal radiation absorption and re-emission by smoke particles — a localized greenhouse effect. This effect manifests as a "radiative cap," where layers of warm smoke inhibit longwave radiative cooling at the surface.
Regions shaded in dark green (Fig. 8 b) correspond to modest negative longwave differences (e.g., –20 to –80 W m⁻²), indicating enhanced downwelling longwave radiation in the presence of fire. This enhancement may result from: (i) the presence of absorbing aerosols (smoke) that re-emit thermal energy; (ii) thermally stratified layers associated with fire-induced heating acting as a radiative lid; or (iii) cloud formation and intensification in the fire simulations — clouds that were absent in the no-fire scenario — potentially promoted by smoke-enhanced convection, as also reported by Menezes et
al. (Menezes, I. C., Lopes, D., Fernandes, A. P., Borrego, C., Viegas, D. X., Miranda, 2024).

Between 15:00 and 16:00 UTC, the shortwave shading caused by smoke particles is more intense and spatially extensive, with localized reductions in solar radiation reaching up to ~500 W m⁻². In contrast, during the 16:00–17:00 UTC interval, the thermal signature of fire is clearly observed in the longwave flux over the centre of the domain. During the nighttime period



(21:00–22:00 UTC), the enhancement in longwave radiation becomes more spatially confined and aligned with regions

exhibiting higher smoke absorption, suggesting intensified radiative trapping by the smoke plume.




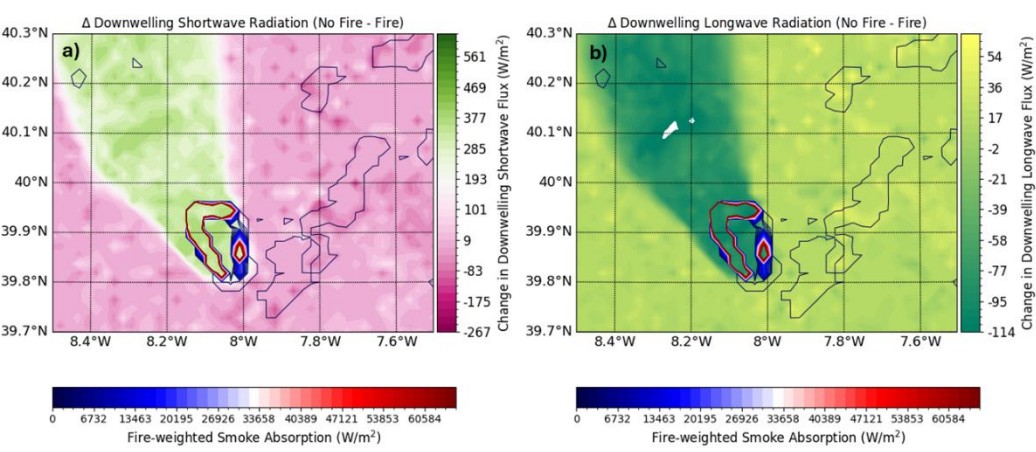

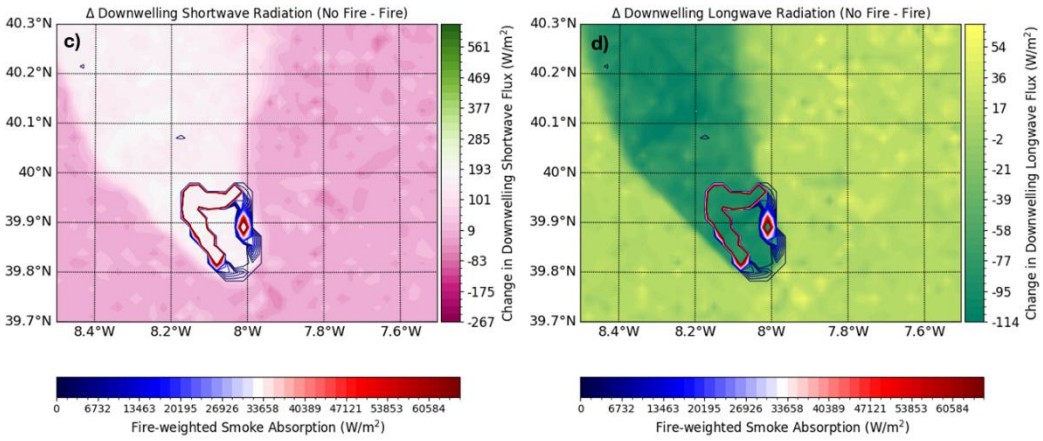

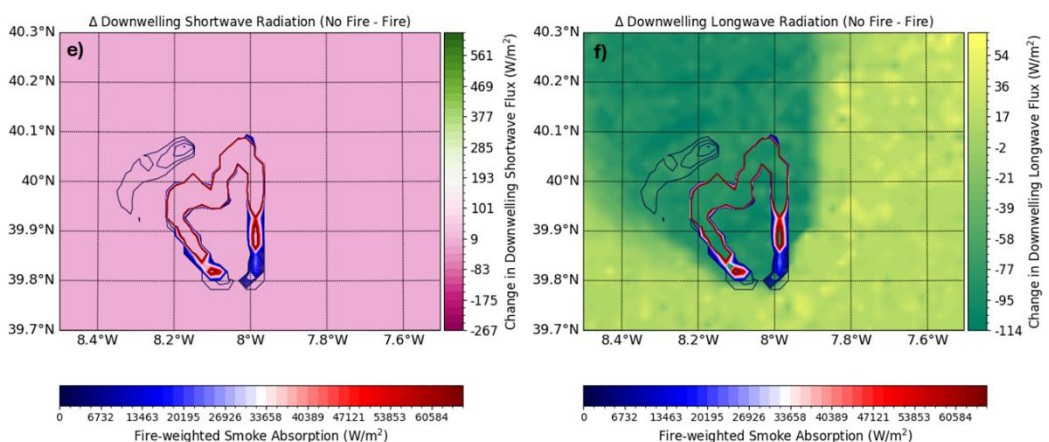



**Figure 8: Difference in shortwave and longwave surface radiative fluxes between simulations with and without fire, at at 16:00,**
**17:00 and 21:00 UTC on 15 October 2017. Positive values in the shortwave and longwave flux difference maps (light pink/light**
**green) indicate a reduction in radiation due to the presence of smoke and fire. Negative longwave values (darker green shades)**
**suggest an increase in downward radiation, potentially associated with thermal re-emission from the smoke plume or cloud**
**formation. Coloured contour lines represent fire-weighted smoke absorption, indicating regions where smoke exerted the strongest**
**radiative influence.**

Fig. 9 shown longitudinal cross-sections of the smoke extinction and absorption coefficients at 550 nm under CAPE and CIN conditions, for 15 October 2017. The colour shading represents the magnitude of the extinction and absorption coefficients, while CAPE and CIN isolines are superimposed: dashed yellow lines indicate the fire simulation and solid orange lines represent the no-fire scenario. Note that the colour bar is nonlinear above ~3 m⁻¹, with wider intervals at higher values, which highlights stronger aerosol loads.

The SOD, inferred from the extinction and absorption coefficients, is primarily confined within the lowest 300–400 m of the atmosphere, where a compact and optically dense smoke plume is observed. This region coincides with maxima in both extinction and absorption, especially between longitudes ~8.1° W and 7.9° W.

Between 17:00 and 21:00 UTC, fire simulations display a consistent upward displacement of CIN isolines compared to the no-fire scenario, particularly between latitudes 39.9° N and 40.0° N. This elevation coincides with the regions of maximum 745 smoke extinction and absorption, suggesting significant local heating by the smoke plume. The thermally induced vertical lifting of the inhibition layer is consistent with radiative forcing due to smoke, particularly evident in the vicinity of the plume core. In several instances, CIN weakens, indicating a lowered convective barrier and facilitating easier access of air parcels to the level of free convection (LFC).

Simultaneously, CAPE isolines over the smoke-affected region exhibit a slight vertical tilt and, in some regions, a local 750 enhancement. This response is most evident between 15:00 and 18:00 UTC, when the plume is most developed, and absorption coefficients exceed 8 m⁻¹. This pattern is consistent with the findings of Menezes et al. (Menezes, I. C., Lopes, D., Fernandes, A. P., Borrego, C., Viegas, D. X., Miranda, 2024), who described a fire-induced thermal low forming within a mountain valley near Sertã. The combination of intense radiative heating from fire combustion and pyrolysis, valley topography, and upslope mountain breeze dynamics likely contributed to mesoscale lifting and pressure anomalies, 755 reinforcing vertical displacement of air layers and reducing local CIN.

These thermodynamic alterations indicate atmospheric column warming and a thickening of the stable layer, supporting the presence of a "radiative lid" that inhibits nighttime radiative cooling and suppresses convective development.

The vertical profiles of SOD—particularly in regions where absorption coefficients exceed 4 m⁻¹—coincide with the steepest vertical gradients in CIN and CAPE isolines. This underscores the pivotal role of aerosol–radiation interactions: enhanced 760 shortwave absorption by BC–rich smoke leads to localized heating aloft, reinforcing thermal stratification. The result is a layered structure where the atmosphere becomes more stable near the surface and more buoyant above the plume. This





feedback loop modifies the vertical distribution of CAPE and elevates CIN layers, as clearly observed in the contour evolution between 15:00 and 21:00 UTC.

Extinction and absorption coefficients are substantially higher at 400 nm—for instance, peak extinction values exceed

67 m⁻¹, whereas in the same region they are around 26 m⁻¹ at 550 nm. This is consistent with the strong interaction of UV– blue radiation with OC aerosols, which are more efficient scatterers at shorter wavelengths. The increase highlights the enhanced sensitivity of the radiative signal at 400 nm to OC-dominated smoke, reflecting the spectral dependence of aerosol–radiation interactions.





**Figure 9: Longitudinal cross-sections of smoke extinction and absorption coefficients at 550 nm and 700 nm along 8.1° W, under CAPE and CIN conditions. Dashed black lines indicate the fire simulation, while dashed orange lines represent the no-fire scenario, at 15:00, 18:00, and 21:00 UTC on 15 October 2017. The region enclosed between the two light gray dashed vertical lines represents the observed area affected by the wildfire.**

Building upon the previously described vertical redistribution of CAPE and CIN in the presence of smoke, further insights emerge from the detailed analysis of thermal inversion structures. The joint evaluation of extinction and absorption



coefficients at 550 nm with potential temperature and relative humidity fields (Fig. 10) reveals that radiative perturbations induced by smoke extend beyond static CIN elevation and CAPE modulation—also influencing the integrity and vertical structure of inversion layers themselves.

In the fire scenario, localized radiative heating beneath the inversion base induces systematic lifting and, in some cases, partial erosion or weakening of the inversion. This effect is particularly pronounced between 17:00 and 18:00 UTC, with displacements of 100–200 m above the no-fire baseline, mirroring the regions where absorption exceeds $4\,\mathrm{m^{-1}}$ and extinction is maximized. The inversion top in these cases becomes less sharply defined, suggesting an incipient destabilization driven by sub-inversion heating.

This vertical restructuring corroborates the presence of a radiative lid, as previously inferred, but adds further resolution to its dynamics: the heating not only shifts the inversion layer upward but also reduces its thermal gradient, facilitating vertical mixing and potentially enhancing entrainment near the plume–inversion interface. These effects are amplified when absorption is concentrated near the inversion base, especially under low-humidity conditions that suppress latent cooling.

Comparative analysis across wavelengths reinforces this interpretation. At 400 nm, the inversion modulation is sharper and more extensive, driven by the stronger interaction of UV–blue radiation with fine-mode aerosols, particularly OC. In contrast, the response at 700 nm is more subdued, with reduced vertical gradients and smaller displacement amplitudes, highlighting the wavelength-selective efficiency of smoke in modifying thermal structure.

Moreover, the co-location of enhanced absorption and inversion erosion suggests a synergistic feedback: as the inversion weakens, vertical transport of aerosols may increase, exposing more absorbing particles to direct solar radiation and sustaining the heating process. This cycle further perturbs boundary-layer stability, contributing to the development of localized thermally driven circulations observed during the late afternoon hours.

Taken together, these results consolidate the hypothesis of a dynamically active smoke layer capable of modifying both static stability and mesoscale vertical structure. By altering the height, strength, and continuity of inversion layers, fire-emitted aerosols reshape the thermodynamic profile of the lower troposphere, reinforcing the broader radiative-dynamic coupling illustrated throughout this analysis.





**Figure 10: Longitudinal cross-sections of inversion layers overlaid with smoke extinction and absorption coefficients at 550 nm and 700 nm along 8.1° W. Dashed red lines indicate relative humidity (%), and solid black lines represent potential temperature (K). Simulations correspond to 15:00, 18:00, and 21:00 UTC on 15 October 2017. The region enclosed between the two light gray dashed vertical lines represents the observed area affected by the wildfire.**



## 4 Conclusion

The results of this study demonstrate that the BRAMS model, enhanced with detailed aerosol–radiation interaction schemes, is capable of simulating the thermodynamic and radiative impacts of wildfire smoke with high spatial and temporal resolution. Simulations of the 15 October 2017 fire event in central Portugal showed that smoke significantly perturbed the atmospheric column by enhancing shortwave absorption (particularly at 400–550-700 nm), leading to the development of thermally induced "radiative lids" and the upward displacement of CIN layers by 100–200 m. CAPE was locally enhanced, and the boundary layer destabilized, revealing consistent radiative-convective feedback mechanisms. Quantitatively, absorption coefficients exceeded $8\,\mathrm{m}^{-1}$ within the plume core at 550 nm, and extinction at 400 nm reached $67\,\mathrm{m}^{-1}$, highlighting the spectral sensitivity of the smoke's optical effects. These findings align well with previous literature identifying OC as the dominant contributor to smoke optical properties and confirm that smoke can significantly influence mesoscale circulations through radiative heating.

The comparison with satellite-derived AOT from MERRA-2 provided further confidence in the model outputs, particularly regarding the spatial extent and spectral behaviour of the plume. The simulated SOD at 550 nm showed excellent agreement with integrated spectral SOD, reinforcing its use as a representative diagnostic of smoke burden. Furthermore, vertical profiles of extinction, absorption, CAPE, CIN, and thermal inversion corroborated the internal consistency of the simulations and their ability to reproduce known fire–atmosphere interaction patterns.

Despite these promising results, the robustness of the model must be further tested across diverse fire regimes. Ongoing efforts aim to validate the framework over a 20-year climatological period encompassing multiple fire events of varying intensities and meteorological conditions. This step is essential to assess model generalizability and to support the development of a predictive tool for operational use. While the current configuration is already well suited for detailed case studies and sensitivity experiments, improvements are still needed in the representation of aerosol aging, plume rise dynamics, and cloud–aerosol interactions under complex topography.

In summary, the BRAMS model with smoke–radiation coupling provides a robust foundation for advancing our understanding of fire-induced perturbations in the atmosphere. Its capacity to reproduce key dynamic and thermodynamic responses to aerosol loading underscores its value as a prognostic tool. Future work should focus on expanding its validation, refining aerosol microphysics, and integrating ensemble capabilities to enhance forecast reliability. These developments are critical for improving fire–climate feedback assessments and for supporting early warning systems in fire-prone regions under a changing climate.

## Code availability

The version of the BRAMS model used to produce the results presented in this paper is available under the Creative Commons Attribution 4.0 International License on https://zenodo.org/records/15830137 (CPTEC/INPE, 2025), along with compilation and simulation instructions.

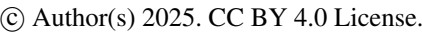



**Author contribution**

ICM and LFR developed the coupling within the BRAMS model. ICM also designed the presented experiments, performed
the validation, produced the figures and analysis, and wrote the original manuscript. KML and SRF guided the
conceptualisation of the coupling implementation in BRAMS. MFF supported the development by correcting conceptual
coding errors and assisting with model compilation. RB, VFO, and SC created supporting files for the simulations. In
addition, SC contributed to the manuscript review. AIM reviewed the manuscript and supervised the conceptualisation of the
model's application.

**Competing interests**

The authors declare that they have no conflicts of interest associated with this publication and that no financial support has
influenced the outcomes of this work.

**Financial support**

The authors acknowledge the financial support of FCT – Science and Technology Portuguese Foundation, which funded the
855 project FIRESMOKE (http://doi.org/10.54499/PTDC/CTA-MET/3392/2020), through national funds. Thanks, are also due
for the financial support given to CESAM by FCT (UID Centro de Estudos do Ambiente e Mar (CESAM) +
LA/P/0094/2020), through national funds.

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
