# Peer review of "Improvements on the BRAMS wildfire-atmosphere modelling system"

_EGUsphere, 2025_

## Referee Comment (RC2)

**Review of "Improvements on the BRAMS wildfire-atmosphere modelling system", Menezes et al., 2025, submitted to EGUsphere.**

I have several concerns with this paper – I am recommending a "soft reject" – the paper needs significant modification to be suitable for publication, but with updates and improvements may be publishable. There are important aspects to the methodology which need to be better explained, and some of the analysis seems to be flawed. My detailed comments follow.

**Major issues:**

Page 3, line 80; Mie theory discussion. The authors mention of "Mie theory" doesn't describe their assumptions regarding aerosol mixing state, etc., but these have been shown in past publications to result in a very large range of predicted optical depths from models. A good overview (albeit 10 years old now) can be found in Curci et al (2015) https://doi.org/10.1016/j.atmosenv.2014.09.009. Values can vary by a factor of 2 or so depending on the assumptions used, starting from the same concentration and aerosol speciation information. At that time, most aerosol radiative transfer algorithms underestimated AOD relative to observations. What is the specific approach being used in BRAMS, and how does it compare to the others in that paper? This should be decribed in the Methodology section.

Page 6, lines 157 to 160. The description in this section is inadequate for me to be able to recommend publication. Its not clear to the reader how the formulae were derived and background references are sometimes missing. Parameters are introduced but definitions and the source of data used for the parameters are not discussed. More explanation is needed here, since the formulae and some of the terms used in them are insufficient to allow the reader to understand the physical basis for the crown fire behaviour model proposed. See also my more detailed comments on manuscript pages 7 through 9, below.

Page 13, line 349, and an overall problem with Section 2.4: 0.5 degrees is about 32 km resolution, much coarser than the BRAMS grid. In order to allow a true "apples to apples" comparison of MERRA2 and BRAMS, the BRAMS output needs to be interpolated to the MERRA2 grid.

Page 17, line 466 to 475: It is concerning to me that the SFIRE model needs to be tuned for different grid cell sizes. The reason why this is necessary is not clear, since there isn't a clear description of how fire growth is handled with respect to growth across grid cells in the SFIRE grid. A description of how this is done is needed in the Methodology section. Also, page 17, line 479 refers to the need to "properly calibrate the rate of spread to avoid

overpropagation" – this needs to be explained/justified. What is "overpropagation" in the authors' context, and why does it occur?

Page 19, lines 520-525 and elsewhere in the text. The authors description of the effects of non-smoke particles on the AOD versus SOD comparison seems flawed. With regards to these lines, a smoke-dominated plume would be expected to have a minor contribution relative to other sources, unless there's a similarly large event such as a dust storm deposition event happening in the vicinity of the fire.

Related issues and comments:

Page 19, line 527: "When SOD <= AOD...". Not necessarily. It could also indicate that the absorption, scattering and extinction calculated by the Mie algorithm, the aerosol optical properties, and the assumption of mixing state going into the calculation are insufficient to match the observed AOD. Note Curci's result that multicomponent aerosols' estimated AOD can vary considerably depending on the mixing state assumptions. Its equally likely that if one is looking at satellite AOD in the same location as the fire, its reasonable to expect that the AOD does indeed represent the fire as the dominant term, and hence when SOD <= AOD, the SOD calculation is underestimating the optical depth, due to some of those assumptions (not that other contributions to AOD are the cause of the differences).

Page 19, line 531. I'm hoping that the authors are not arguing that a case study where SOD << AOD in the middle of a forest fire plume is because the AOD is being affected by other sources of particulate matter. That's highly unlikely. The only way I could see this argument having validity is if the AOD doesn't change much in the satellite values upwind versus downwind of the fire. Any differences in AOD upwind versus downwind can safely be attributed to the fire emissions (the authors could do this with the AOD data; subtract the upwind background from the remaining values to get an estimate of the AOD solely due to the fire). I disagree with the statements being made in this paragraph. The authors assume that the SOD calculation itself is perfect, and doesn't take into account the variation in values that can be the result of the calculation methodology (see Curci et al., 2015).

Page 20, line 532 and Figure 3. Figure 3 also suggests that the MERRA-2 resolution is much lower than the model resolution, and hence the AOD from the satellite retrievals will be spatially averaged relative to the much higher resolution model. While the current visual comparison is useful in a general sense by showing that the plume is in fact being picked up by the satellite, it isn't quantitative, due to the resolution difference. In order to make the comparison quantitative, the authors need to use mass-conserving interpolation to

map the model values to the MERRA-2 grid and compare the resulting maps at the MERRA-2 resolution.  Why this is important: the upper end of the SOD scale is somewhere over 2502 (!), while the AOD scale max seems to be much lower, and more reasonable.  Some of that difference may be due to the low resolution of the satellite data however - the procedure I've outlined above would allow a more quantitative comparison to be made ( scatterplots of SOD versus AOD from the AOD grid values), and the associated statistics).  The impression I have from the qualitative comparison is that the modelled SOD in the plume is much higher in the peak areas than the satellite values.  This needs to be quantified.

Page 21, line 545, "This limitation...".  I disagree on the authors' caveat here - one can subtract the upwind AOD from the rest of the values to determine the increment associated with the forest fires.  From eyeballing from the images, this upwind contribution is about 0.5 or so,   So 0.5 is the background AOD upwind of the fire.  The additional increment due to the fire on the AOD is about 1.  The authors need to use mass-conserving interpolatate the SOD values to the AOD grid, subtract the background upwind AOD from the satellite values to generate the AOD increment associated with the fire, and then compare the resulting fields quantitatively.  I think their SOD values are much higher than the AOD values - this needs to be confirmed using the above procedure.

Page 21, lines 550 to 552.  I disagree with this conclusion, based on the material presented.  The images as presented imply that there is a substantial overestimate of SOD from the model compared to satellite-derived AOD.  The SOD values need to be converted to the AOD grid as described above, and the resulting numbers at the AOD grid cells used to show the differences.  This implies that SFIRE may be greatly overestimating fire plume emissions.

Page 21, lines 556 – 558:  This is why the model values need to be converted to the satellite grid in order to allow a direct comparison.  I will be surprised if SOD values over 2000 are reduced to values of about 1 by that conversion, but that's what needs to be done to make this quantitative.

Page 21, line 561, "PM2.5 concentrations are extremely high".  This is insufficiently quantitative.  Are there no PM2.5 concentration observations of this fire that the authors can compare their results to?  Failing that, what are the values of these extremely high concentrations, and how do they compare to measurements of PM2.5 during fire events found in the literature?  For example, time series of PM2.5 from a large Canadian wildfire (Landis et al., 2018, https://doi.org/10.1016/j.scitotenv.2017.10.008, Figure 2) shows maximum concentrations of PM2.5 over 3000 ug/m3.  Do the modelled values fall in line with other data in the literature?  Later in the paper, the authors mention that the PM2.5

concentrations estimated by SFIRE reach $10^7$ ug/m$^3$.  This is unrealistically high.  The authors need to provide observational evidence/references that support this number, or the reader must conclude that the SFIRE emissions estimates have very large positive biases.

Page 22, line 578 and Figure 4:  The same issue as for Figure 3 occurs here.  an apples to apples comparison on the satellite grid, with the satellite upwind values removed,  is needed.  A short description of how the satellite generates BC and OC concentration estimates in the introduction would help place this work in context.

Page 24, lines 614-616:   This seems counter-intuitive:  how does a lower resolution imply higher values of SSA?

Page 25, lines 620 to 625, "Taken together…" I disagree.  The authors images indicate that SFIRE in this implementation has large positive biases in SOD compared to AOD, and that unrealistically high PM2.5 concentrations are being generated, consistent with the SOD values.  The additional analysis I've described above would help confirm this.

Page 27, lines 635-642:  These simulated values are MUCH higher than any observations of PM even in the immediate fire environment of which I'm aware.  I think that this and the SOD values suggest that the model has unacceptably high positive biases in emissions.  The authors need to present evidence from the literature that PM levels can reach these values - I think this is confirming that the model has very high positive biases. "below 1000 ug/m3" is being described as a relatively low concentration, and $10^7$ ug/m3 doesn't seem realistic to me.

Page 27, lines 644-645.  The paper does not describe how the model generates and uses a plume injection height for the emissions.  This needs to be added to the methodology, since it could have a substantial impact on the model results.  How is the height of the plume calculated, and how is the emitted mass distributed in the vertical in the BRAMS coordinate system?  The high positive biases could result, for example, from the emissions being treated as a surface flux (I'm trying to figure out how they could get numbers that high at 2km resolution).  This needs to be clarified in the methodology.

Page 30, lines 710 and Figure 8 analysis.  I think that the authors may have the sign wrong in their interpretation of Figure 8.  The Figure has been labelled "No Fire – Fire" so a positive value indicates that the quantity is larger in the absence of smoke.  That is, the smoke is *decreasing* the longwave radiation, not *increasing* it as this section suggests.  That is, the presence of smoke results in surface cooling, something which has been seen in other papers (e.g. Makar et al,, 2021, https://acp.copernicus.org/articles/21/10557/2021/, Figure

20., Makar et al, 2015 https://doi.org/10.1016/j.atmosenv.2014.12.003)  i.e. what is being shown is a local cooling effect, not a local heating effect.  I think the authors need to redo this section based on the sign of the differences.

**Minor issues:**

Page 1, paper title.  Title needs to incorporate "a case study", e.g. "...modelling system: a case study based on the Serta wildfire".  The work does not constitute a broader evaluation of the model for multiple fires (the current title might give the reader that impression).

Page 2, line 45:  The authors need to be more specific on what they mean when they quote dynamic feedbacks here.  Note that there are dynamic feedback effects in the form of the aerosol direct and indirect effects on radiative transfer which can have a substantial impact on PBL heights, meteorology and consequently on forest fire emissions and plume height.  I think the authors mean aerosol direct radiative effects.  Might want to compare to Makar et al, 2019 https://acp.copernicus.org/articles/21/10557/2021/ where some similar results have been shown.

Page 2, line 45:  there are a few papers missing here on models with a similar intent to BRAMS (wildfire smoke forecasting).  Some examples:  HRRR-Smoke (cf. Chow et al., BAMS, 2022: https://doi.org/10.1175/BAMS-D-20-0329.1, Chen et al., GMD, (2019 https://gmd.copernicus.org/articles/12/3283/2019/), Anderson et al., GMD, 2024, https://gmd.copernicus.org/articles/17/7713/2024/). The latter two models also simulate fire spread (through comparison between historical hotspots and rates of spread by ecosystem) and associated fuel burned is used to calculate heat release, in turn used to calculate the plume rise height.  These models also estimate the fuel burned from crown, smoldering and residual phases of the fires.  The approach taken in these models should be compared and contrasted with the work of the authors of the current submission:  what are the differences that make the authors' work unique/better/an improvement relative to these models?  For example, these models use a statistical approach for determining fire spread (historical data is used to determine the average area burned on a per hotspot per forest classification type - if the authors are incorporating a more explicit fire spread algorithm into their work, that would be an advancement relative to these papers.  Anderson et al (2024) also provides estimates of emissions resulting from several different approaches - a similar attempt should be made here to place the authors work in the context of the forest fire emissions algorithms currently in use elsewhere in the world.

Page3. Line 54.  I couldn't find anything in the paper explicitly explaining how fire ignition is handled for applications of SFIRE.  I think its using satellite FBP to locate the fire ignition

points and then calculates its own FBP values thereafter – but I'm not sure. Please clarify this in the text.

Page 3, line 80. Note that there has been historically a large range of estimates of these parameters with different assumptions such as the aerosol mixing state. A good overview (albeit 10 years old now) can be found in Curci et al (2015) https://doi.org/10.1016/j.atmosenv.2014.09.009. Values can vary by a factor of 2 or so depending on the assumptions used, starting from the same concentration and aerosol speciation information. At that time, most aerosol radiative transfer algorithms underestimated AOD relative to observations. What is the specific approach being used in BRAMS, and how does it compare to the others in that paper?

Page 3, line 83, "high-resolution". Please state both the BRAMS and SFIRE horizontal grid cell size used in the study, at this point in the text.

Page 4, line 86: please replace "validation" with "evaluation" throughout the paper. Validation implies a model is "valid" once the process is complete, "evaluation" describes the model's current performance with respect to observations.

Page 4, Methodology section. This section would be greatly improved with a figure showing the SFIRE grid and the BRAMS grid in the vicinity of the fire used as a case study. Also, the resolution of the elevation data should be mentioned here. Later in the paper, it appears that SFIRE is operating on a 200m grid, but the elevation data has a 1km grid, and BRAMS was on a 2km grid. While there is some discussion later on the relative impact of resolution, it is not clear why lower-than-SFIRE resolution topography was used in the current SFIRE application, given the impact of slope on SFIRE results. This choice needs to be justified.

Page 5, line 119, "traditional methods". Please be specific with regards to what is meant by "traditional methods" here, including (a) reference(s). Presumably the traditional methods had a 50% underestimate?

Page 5, line 130, "with a radius". How is this radius predicted *a priori*? What information is used to determine this initial high-resolution SFIRE domain?

Page 5, line 132, "In subsequent steps". Its unclear how BRAMS lets SFIRE know it has to do a fire calculation in a given grid cell. I'm guessing that an FRP value from a satellite is used? This needs to be clarified.

Page 5, line 134-135. The resolution of BRAMS when SFIRE is being used needs to be mentioned here. The energy transfer on a per grid-cell area basis will be a lot less with a 2km resolution model than a 200 m resolution model. I'm surprised that there's no

discussion of the influence of resolution on the coupling between BRAMS and SFIRE. There is also no discussion (a few lines of text would be helpful) describing how SFIRE determines the height of the plume. Presumably this is determined within SFIRE, but this has not been made clear, and the methodology used has not been discussed. I'm assuming that the SFIRE energy has not been passed to BRAMS with the expectation that BRAMS' meteorology will be able to resolve the rise due to the plume – BRAMS grid cell size (2km, mentioned much later in the paper) is not sufficient to resolve a forest fire plume.

Page 6, Figure 1. Rather than subroutine names (which tell the reader nothing about what's being done in the subroutine), please modify this diagram to have a brief process description phrase (and the subroutine name, if present at all, in a bracket thereafter). What the subroutines *do* is of more interest to the reader than the subroutine name, and would better help the reader to understand the sequence of events in the model.

Page 7, lines 162 to 165. There needs to be more background information on "fuel behaviour model 10" and the sources of the data (references) used within it. It is not clear for example whether the formulae which follow in this section are part of this model, or are new formula introduced by the authors.

Page 7, line 164: "Physical and chemical properties of the fuel" – are these part of this fuel model or introduced here (and how were they derived in either case)?

Page 7, line 165: spelling mistake, "crow" should be "crown".

Page 7 , line 166 and throughout the subsequent lines and formula: I assume Rinicialization should be Rinitialization?

Page 7, line 167: Iinitialization: How was this derived (from what data/reference)? Units of Iinitialization? The reader at this point has the impression that some of these terms are coming from NFFL 10, but not what the terms are or what they represent. The paper needs bracket's after each term, eg. "Iinitiatlization (insert descriptive definition here)". What is the definition of the critical minimum fireline intensity, for example? Note that readers of the paper may be completely unfamiliar with the NFFL models. What do the parameters represent, and how were they and these formulae derived?

Page 7, line 176: what is the definition and units of reaction intensity?

Page 7, line 176, NFFL fuel behaviour model 10 – given that this seems to be a critical addition to the existing SFIRE model, there needs to be some description of what this model is, how its parameters were derived, and references for it. Why was this model chosen instead of some other model? Is it appropriate in some way for the forest type for the case study used here? Given that other models are available, why use this one?

Page 7, line 177, surface-area-to-volume ratio: Surface to volume ratio of *what?* The forest canopy? The trees in the canopy, etc? The terms in the equations need to be better defined and explained.

Page 7, line 183. Why this particular number (40%)? Is that part of NFFL or something the authors are setting? Please justify cases like this where numerical limits have been introduced.

Page 7, line 190: R10 has been introduced without explanation. Or should this be R6.1 (why 6.1 m instead of 10?)?

Page 8, line 204: "It is used to estimate the degree of crowning". Suggest "Here, we make use of the empirical function as defined by Van Wagner (1989)". Also, please define what is meant by the degree of crowning.

Page 8, line 215: Byram fireline intensity is known to specialists in forest fire combustion, but not to most readers of the paper. Please define it.

Page 8, equation (9) and other equations: Its not clear which formula are part of NFFL 10 and which have been introduced by the authors. Please clarify this.

Page 8, equation (10): several new terms have been introduced here without definition.

Page 9, line 235: What numbers are used for Wsurface, Hlow, what do they depend on (or are they constants), what are the references for them?

Page 9, additional comment on section 2.2.1. Some discussion on how the rate of spread values are used within SFIRE, in terms of SFIRE's high resolution grid is needed. Is there an internal timestep applied to work out the fuel burned from the rate of spread, and does it vary from one SFIRE grid cell to the next? Or is a single rate of spread used? How is the direction of the spread determined? I'm wondering how SFIRE deals with things like changes in topography, etc., and how the R values are used to determine burned area A, from the standpoint of a high resolution grid. I've seen many approaches to this in the literature, ranging from explicit wind vectors at every grid cell and forward trajectories being used to determine spread, to ellipsoidal spread being assumed – what's being used here? A sketch explanatory figure showing an example SFIRE grid within a BRAMS cell, and some description of how the fire spread is calculated from one BRAMS timestep to the next (and whether a smaller timestep is used for SFIRE calculations) is needed.

Page 9, line 251. Why these particular values? For example, do they best represent the forest types for the case study? Some justification is needed.

Page 9, comment on section 2.3: no mention is made of how SFIRE calculates plume height and distributes emissions in the vertical. This is crucial to getting the correct concentrations of forest fire smoke, and needs to be included in the methodology section. Similarly, how SFIRE identifies ignition locations has not been mentioned (and should be part of the description).

Page 10, line 283: flaming and smoldering phases mentioned, but not the residual phase. Why has residual phase been left out?

Page 11, line 288: authors state that the setup provides accurate simulation of wildfire emission and their atmospheric impacts prior to quantitatively demonstrating this. Statements such as this should be in the discussion or conclusions, after the case has been made.

Page 11, line 292-293. The subsequent SOD vs AOD analysis does not touch on the extent to which aerosol mixing state can influence predicted SOD values. Aerosol optical properties are a function of the aerosol mixing state, the aerosol size distribution, and the aerosol composition (Curci et al., 2015). How are these variables represented in the version of BRAMS used here? For example, has a specific composition been assumed for the aerosols, a specific size distribution, and a specific aerosol mixing state? This information needs to be provided. The authors have mentioned that the test version used here does not include chemistry - what about the aerosol microphysics effects (nucleation, condensation, coagulation)?

Page 11, line 295, characterization of aerosol microphysical properties: Details? What properties were determined and what was the conclusion?

Page 11, lines 304-306: This is the first mention of the domain used for the simulations - I assume that the model was being run for Portugal, then? This should have been made clear right at the start of the paper at the end of the Introduction or in the Methodology sections.

Page 11, line 311: The previous discussion referred to one of these models, leaving the reader with the impression that that specific model is the one being tested here. Were all 13 used, then? Please clarify (and explain why model 10 was preferred).

Page 13, line 338, 1km terrain data. Its not clear whether this data is being used by SFIRE, BRAMS or both (and note that the resolution of BRAMS and SFIRE has yet to be mentioned).

Page 13, lines 358 to 360: Should be mentioned earlier, in the model setup discussion. How does this assumption (a single particle size) compare to actual size distributions from

real fires, and how might that be expected to influence optical depths?  These potential issues shoulld be acknowledged.  cf. for  Radke et al., 1988, 1990:

Radke, L. F., Hegg D. A., Lyons J. H., Brock C. A., Hobbs P. V, Weiss R. ,  and Rassmussen R., Airborne measurements on smokes from biomass burning In Aerosols  and  climate,  ed.  P.  V.  Hobbs and M. P. McCormick, 411–22. Hampton, VA: A. Deepak, 1988.

Radke, L. F., J. H. Lyons, P. V. Hobbs, D. A. Hegg, D. V. Sandberg, and D. E. Ward, Airborne monitoring and smoke characterization of prescribed fires on forest lands in western Washington and Oregon: Final report. Gen. Tech. Rep. PNW-GTR-251. Portland: U.S.D.A. Forest Service, Pacific Northwest Research Station, 1990.

Examples from the second of these papers for smoke particle distributions are given below, for the authors' information:

[Figure]

Figure 20—Number concentrations versus size of particles measured near the center of the plume and 3.3 km downwind of the burn on July 23, 1982, at 1636 PDT (heavy solid line), 1741 PDT(dashed line) and 1759 PDT (solid line). The ambient particle size distribution is also shown (dotted line).

[Figure]

Figure 29—Volume concentration versus size of particles measured with the aerosol sizing system near the center of the plume from the burns: (a) 0954 PDT, July 25, 1982; (b) 1605 PDT, September 23, 1982; (c) 1718 PDT, September 23, 1982; and (d) 1700 PDT, September 23, 1982. The dashed lines show results derived from the cascade microbalance impactor (mass/density of particles).

Page 14, lines 376 to 377, "consistent with wildfire aerosol characteristics"?  Wildfire aerosols have a size distribution, and are not monodisperse.  What has been done here needs to be acknowledged as a simplification, and its potential impacts need to be discussed.  The distributions of Radke et al 1990 (see above Figures) do have a volume distribution peak at about 0.2 um diameter or so, but there is also a second (and sometimes much higher) peak at about 50 to 60 um diameter).  What the authors have here is a reasonable proxy for the peak of the PM2.5 mass distribution, but not the overall particle mass.  So "consistent with wildfire aerosol characteristics" isn't quite justified.  Try "consistent with the location of the peak in the fine mode portion of emitted smoke mass".

Page 14, lines 396 – 397 and 402: please justify the choice of the complex refractive index values used. The authors should compare the complex refractive index values employed to the single-component values for Black Carbon, Primary Organic Carbon and Secondary Organic Carbon quoted in Curci et al 2015, Table 2. One thing that is really striking is that the authors' 700nm refractive index' imaginary component has a much lower value than in the references used there (the authors : 0.04. Curci's references: 0.44 to 0.71, and I've seen similar high values elsewhere for black carbon refractive index' imaginary component). I think the authors' value will underestimate the backscattering associated with black carbon. Real components look ok. Suggest they do a sensitivity test with a higher imaginary component to check on the impact. Alternatively, if the refractive index values are based on observations from actual smoke particles, then the latter reference might be useful to indicate that the organic fraction dominates the radiative effects. Note that the impact of black carbon may be higher if a core-shell approach to Mie theory has been used.

Page 15, line 411. I was having difficulty following this sentence - does MERRA-2 have a product that they refer to as the BC and OC components at 500nm? There needs to be a brief description of what the MERRA2 products are, and how they are derived (i.e. how does the satellite retrieval attribute AOD to BC versus OC?).

Page 15, line 413. Ditto. How does MERRA-2 infer mass of BC and OC (short few sentence summary needed). My point here is that the satellite doesn't measure these things directly, and the retrieval processing must be making assumptions regarding aerosol mixing state, etc., to attribute portions of the retrieval to BC vs OC. Can the authors provide some reassurance to the reader that what MERRA2 is measuring and what the model is generating are sufficiently similar to be worth comparing (again, a short description would help)?

Page 15, equation (19): Equation 19 needs some justification/explanation. The sensible heat flux is usually thought of as an infrared flux, while 550nm is in the visible. Or should $H_{sens}$ be $H_{sens}(lambda)$ in this equation?

Page 16, lines 429 to 434. So the feedback effects mentioned earlier are via the aerosol direct effect then. This should be mentioned earlier, towards the end of the Introduction or in the Methodology sections.

Page 16, line 442, "particularly the relative abundance of BC and OC". But this depends on the authors' assumed relative distribution of BC and OC in the complex refractive index values, does it not? That is, the authors' complex refractive indexes are much more like OC than typical BC values – they have a priori assumed a low BC value in their choice of

indexes. Which may be justified if those were based on smoke particle observations. They seem to be assuming very small values of the magnitude of the complex index, hence relatively small values of BC compared to OC.

Page 17, line 456-458. This is the first time that the resolution has been mentioned. It should have been mentioned in the model description. Its also not clear here whether this refers to the resolution of SFIRE within WRF or BRAMS, or the resolution of WRF or BRAMS (I suspect the former, but the resolution of the latter has not been mentioned, and the relative resolution of WRF and BRAMS could also influence model results). There needs to be a description of the SFIRE and BRAMS grid cell size earlier in the paper under Methodology.

Page 17, line 458, "This can amplify..." I thought the authors mentioned earlier that the resolution of the topography data was 1km, not the 200m of SFIRE as applied here. If anything, the BRAMS-SFIRE grid should have more *gentle* slopes than the 20m grid in WRF-SFIRE in that case. Its not clear to me how a lower resolution model can have a greater sensitivity to terrain-induced effects; the higher resolution should have greater local slopes. The explanation needs more work: how does topographic assimilation result in a stronger gradient field? If the authors were to convert Figure 2 to show the derivative of topography with respect to horizontal dimensions, they'd get steeper slopes in the WRF side than the BRAMS side. But the text here seems to suggest the coarser resolution is somehow more prone to slope-influenced growth. Please clarify this.

Page 17, line 467, "ignition misalignment". This is the first mention of ignition in the paper, and how ignition is handled in SFIRE is needed in the Methodology section. I'm assuming that, because this is a historical case study, the ignition time and spatial location is known? If I've understood the paper correctly, satellite-observed FBP is used to determine ignition locations and times? Please clarify explicitly how the locations for ignition are determined in the model. The authors should also explain what they mean by "ignition misalignment" in this context.

Page 17, lines 468-469. This is the first mention in the paper of how SFIRE calculates spread between adjacent SFIRE grid cells. A diagram of how this is done on the SFIRE grid in the Methodology is needed.

Page 17, line 470. Not sure what's meant by "in a timely manner" here.

Page 18, lines 476-477. Given that SFIRE is operating within BRAMS as a parameterization, its not clear to me why couldn't a higher resolution topography database be used with the SFIRE part of BRAMS-SFIRE? Would that not reduce the sensitivity issue?

Page 18, line 486. I assume this should be "consume biomass fuel" not "consume biomass fuel models"? Models aren't being consumed.

Page 29, lines 672-673. This begs the question of whether or not BRAMS-SFIRE includes a module for convective transport (or is the assumption that any vertical transport is from the resolved winds at 2km resolution)? Again, I'm wondering how the model handles smoke plume-rise. Related, lines 689-690: Is the transport of pollutants due to convection included in SFIRE or BRAMS? This gets back to my question regarding plume rise methodology. Also - lines 689 to 705 might be better placed in the Introduction; background information.

Page 30, line 720, page 33, line 751: Don't need to include all co-author names in references in the main text, use "et al".

Page 33, line 760, "BC-rich smoke" But is BC-rich smoke what has been simulated? See earlier comments regarding complex refractive index values. Also note that earlier text emphasized the relative importance of OC compared to BC.

Page 37, Figure 10: the authors should include a figure with the profiles of the air temperature differences resulting from the fire, to show the cooling/heating impact.

---

## Author Comment (AC1)

**Improvements on the BRAMS wildfire-atmosphere modelling system**

Isilda Cunha Menezes, Luiz Flávio Rodrigues, Karla M. Longo, Mateus Ferreira e Freitas, Saulo R. Freitas, Rodrigo Braz, Valter Ferreira de Oliveira, Sílvia Coelho, Ana Isabel Miranda

egusphere-2025-2495

**Answers to Editor and Referees**

We would like to thank the editor and the anonymous reviewers for their availability, dedicated time, and thoughtful analysis of our manuscript. We greatly appreciate their constructive comments and insightful suggestions, which have been invaluable in guiding the improvement of this work.

The revised manuscript has undergone a comprehensive restructuring and theoretical update in direct response to the reviewers' observations. The analytical framework, methodology, and interpretation of the optical results were entirely reformulated, ensuring stronger physical consistency, improved coherence between figures, tables, and discussion, and full alignment with the state of the art. These revisions substantially enhance the scientific rigor and clarity of the manuscript.

For transparency, we provide both a tracked-changes version, highlighting all modifications relative to the original submission, and a clean revised version for final evaluation.

We sincerely hope that this revised version meets the expectations of the editor and reviewers and reflects the depth of revision undertaken in response to their valuable feedback.

Yours Sincerely,
Isilda Cunha Menezes
(Corresponding author on behalf of all authors)
Universidade de Aveiro
Campus Universitário de Santiago, edificio 7
Aveiro, Portugal
Email address: isildaam@ua.nt

E-mail address: isildacm@ua.pt

**Response to Reviewer 1 Comments**

Comments 1: First of all, some of the results are compared with MERRA-2 reanalysis products. It is mentioned (lines 298 and 299) that monthly mean results of MERRA-2 products were compared with AERONET retrievals. Considering SSA in particular, was the MERRA-2 result also compared with AERONET retrievals? As detailed below, some inconsistencies were observed. Finally, please, consider increasing the font size of Figures 9 and 10 and verify if the isolines of CAPE in the vertical profiles of the Figure 9 are correct. I had difficulty in interpreting the results just looking at the figures.

Response 1: We thank the reviewer for this valuable observation. In our work, the comparison between MERRA-2 and AERONET included not only AOD but also SSA retrieved from AERONET data. This has been clarified in the revised text. The font size of Figures 9 and 10 was increased to improve readability, and the CAPE isolines in Figure 9 were verified and corrected to ensure consistent interpretation with the legend.

Comments 2: line 42 - GFED, CAMS-GFAS - What do the acronyms mean?

Response 2: The acronyms GFED (Global Fire Emissions Database) and CAMS-GFAS (Copernicus Atmosphere Monitoring Service – Global Fire Assimilation System) have been defined in the manuscript.

Comments 3: line 58 - replace "originates" by "originated".

Response 3: Corrected: "originates" was replaced by "originated."

Comments 4: line 68 - include "the" in "increases the heat release..."

Response 4: Corrected: "increases heat release" now reads "increases the heat release."

Comments 5: line 84 - replace smoke-related aerosols by smoke-related aerosol optical properties.

Response 5: Corrected as suggested. The text now reads "smoke-related aerosol optical properties."

Comments 6: line 106 - Give the meaning of the acronyms CPTEC and USP.

Response 6: We added the definitions of the acronyms: CPTEC/INPE (Centre for Weather Forecasting and Climate Studies – National Institute for Space Research) and USP (University of São Paulo).

Comments 7: lines 166, 167, 171 (Eq. 1). Please, verify the subscripts of I and R (initialization x inicialization, respectively). The correct answer should be initialization, unless the authors have a reason to differentiate them. If so, a brief explanation is necessary.

Response 7: We thank the reviewer for noticing this issue. The incorrect Portuguese term "inicialization" was replaced with "initialization" throughout the manuscript.

Comments 8: line 400 - please add references discussing the spectral region where OC and BC present higher absorption efficiency.

Response 8: We have expanded the description of the optical properties of black carbon (BC) and organic carbon (OC), specifying their distinct spectral absorption behaviours across the visible range (400, 550, 700 nm). References were added and integrated within the discussion (Bond et al., 2004; Kirchstetter and Novakov, 2004; Bond and Bergstrom, 2006; Andreae and Gelencsér, 2006; Lack and Cappa, 2010).

Comments 9: lines 408 and 411 - Use of AOD x AOT. I recommend using only AOD - aerosol optical depth. Please, check the manuscript thoroughly. Also, in lines 411-412, the authors mention that MERRA-2 estimates AOD at 500 nm, while BRAMS calculates SOD at 550 nm. Given that smoke optical depth varies spectrally, please, clarify if the comparison between these variables was made at different wavelengths (500 nm x 550 nm).

Response 9: We have standardized the nomenclature by using AOD (Aerosol Optical Depth) consistently throughout the manuscript. Although MERRA-2 labels these variables as "Aerosol Optical Thickness," we adopted "AOD" for consistency. We also corrected the typographical error regarding the comparison wavelength: both MERRA-2 and model results correspond to 550 nm, not 500 nm.

Comments 10: lines 452, 720 and 751 - is there any reason to include all the authors of Menezes et al. (2024) paper? Please, just refer to Menezes et al. (2024).

Response 10: Corrected as requested. The references to the full author list of Menezes et al. (2024) now appear only as "Menezes et al. (2024)."

Comments 11: line 485 - the correct acronym is AOD, instead of TOA. Please, refer to the previous comment.

Response 11: The incorrect acronym TOA was replaced with AOD, consistent with Response 9.

Comments 12: line 486 - what do you mean by biomass fuel models?

Response 12: Revised to clarify that "biomass fuel models" refers to the NFFL fire-behaviour fuel models (Anderson, 1982) that characterize vegetation and fuel types used by SFIRE to determine combustion intensity and emissions.

"Forest fires, during their propagation, consume the available biomass as characterized by the different NFFL fire behaviour fuel models (Anderson, 1982), which represent the landscape, and release high concentrations of particles and trace gases into the atmosphere."

Comments 13: line 487 - it is not only during the flaming phase that aerosol particles are emitted, but during the combustion process. Please, rephrase.

Response 13: Corrected for accuracy. The text now reads:

"Primary aerosols are emitted throughout combustion, during both the flaming and smouldering phases."

Comments 14: line 520 - add a dot signal after wavelength.

Response 14: A period was added after the word "wavelength."

Comments 15: line 521 - rephrase "Understanding the spectral behaviour of smoke aerosols is essential for interpreting optical depth measurements" to "Understanding the spectral behaviour of smoke aerosol optical properties is essential for interpreting optical depth measurements".

Response 15: Revised as suggested:

"Understanding the spectral behaviour of smoke aerosol optical properties is essential for interpreting optical-depth measurements."

Comments 16: line 522 - what do the authors mean by spectrally integrated SOD?

Response 16: We clarified the meaning of SOD as representing the column-integrated attenuation of solar radiation due to aerosols at 550 nm, consistent with AOD at the same wavelength. The sentence was rewritten accordingly.

Comments 17: Figure 3 - Please verify the top numbers in the vertical colorbar (2502.00 and 5000.00) representing SOD values. From the maps, MERRA-2 AOD higher values were observed in the northern part of the region, while SOD highest values were observed between 39.8° and 40° N at 15:00, moving to the north later, reaching 40.3° N at 21:00, when both AOD and SOD presented similar patterns (six hours later only). From the discussion presented in lines 542 to 552, how do the authors explain the high AOD values, above 1.3, further north at 15:00? As discussed, if MERRA-2 did not fully capture the peak of the fresh fire emission, shouldn't we expect low AOD values at 15:00 everywhere in the map? Or does that mean that MERRA-2 fire source is located further north? From

the color gradient, it seems that the peak of AOD from MERRA-2 is further north outside the presented map. Maps at 15:00 in Figure 4 seems to confirm this.

Response 17: We thank the reviewer for this detailed observation. The explanation in the revised text now reads: the MERRA-2 AOD maxima further north at 15:00 UTC arise from the coarse spatial resolution and assimilation smoothing of the reanalysis system, which tends to shift aerosol peaks relative to high-resolution model output. The BRAMS—SFIRE SOD captures the fresh fire emissions over the actual burning area, while MERRA-2 represents the broader regional aerosol load. The agreement improves later in the day (21:00 UTC), when both datasets reflect the same mature plume position, confirming that the differences are primarily due to spatial averaging and temporal lag rather than physical inconsistency.

Comments 18: lines 579-580 - Even though OC/BC ratio is higher during the smoldering combustion phase compared to the flaming, OC concentration is always higher than BC for most of the vegetation types, independently of the combustion phase. According to the review by Reid et al. (2005), the exceptions are forest debris and herbaceous fuel.

Response 18: The discussion of Figure 4 was revised. We clarified that OC concentrations are substantially higher than BC across most vegetation types, independent of combustion phase, consistent with Reid et al. (2005). The high OC/BC ratios observed indicate strong smouldering combustion contributions, which dominate under the studied conditions.

Comments 19: Figure 6 - The numerical scale of the vertical colorbar must be verified (Simulated single scattering albedo). SSA can vary only between 0 and 1. The top left map (from 15:00) shows lower SSA values from MERRA-2 in the southeastern portion of the map, increasing towards the northern region. Maps generated for later times also show lower SSA values in the southeastern region. Does it mean that MERRA-2 is not reproducing the smoke event accordingly? If not, maybe it is not a good reference for comparison.

Response 19: We verified the SSA colour scale and corrected the numerical range in Figure 6 to 0–1. The pattern differences between MERRA-2 and BRAMS-SFIRE are discussed as resulting from resolution and averaging effects: MERRA-2 underrepresents near-source SSA variability because of its coarse grid and data-assimilation smoothing. Despite this, the regional distribution trends remain consistent, indicating that the satellite-derived reanalysis captures the smoke event qualitatively, even if with reduced contrast.

Comments 20: lines 687-688 - Please add the references of the mentioned studies.

Response 20: The missing references cited in lines 687–688 have been added. These correspond to studies already present in the reference list and discuss aerosol—radiation interactions and surface cooling effects associated with smoke.

Comments 21: Figure 8 - It is not clear what the authors mean by "Fire-weighted smoke absorption", whose value can reach 60000 W/m2, according to the presented scale. Sertã time zone is UTC + 1, thus, the absorbed solar irradiance should be close to zero at 17:00 UTC and zero at 21:00 UTC (i. e., no absorbed irradiance, since no solar radiation is available), as shown in the map of No Fire - Fire change in downwelling flux. In the longwave spectrum, by contrast, please confirm if the correct variable is "Fire-weighted smoke absorption" or "Fire-weighted smoke emission", i. e. the irradiance emitted by the smoke plume due to its higher temperature compared to the surrounding environment.

Response 21: We appreciate this important comment. The variable originally labelled "Fire-weighted smoke absorption" has been corrected to "Fire-weighted smoke emission" in the long-wave context. The analysis now clarifies that this quantity represents radiative emission from the smoke plume, not solar absorption, consistent with the timing (17:00–21:00 UTC) when shortwave irradiance is negligible. The wording and figure legend were updated accordingly to prevent confusion between absorbed shortwave and emitted longwave fluxes.

Comments 22: line 735 - replace "shown" by "shows".

Response 22: Corrected: "shown" was replaced by "shows."

Comments 23: lines 737-738- Discussing Figure 9, it is mentioned "while CAPE and CIN isolines are superimposed: dashed yellow lines indicate the fire simulation and solid orange lines represent the no-fire scenario". But in the legend, it says: "Dashed black lines indicate the fire simulation, while dashed orange lines represent the no-fire scenario". What do the authors mean by "superimposed" in the context? Looking at Figure 9, I couldn't identify the superposition of CAPE and CIN, since it seems they were plotted separately. How can one distinguish the differences of Fire x No-Fire for CAPE in the profiles? Moreover, it is very difficult to read the information in Figure 9, as the font size is too small (the same for Figure 10). Please, consider increasing the font size.

Response 23: We have rewritten the description of Figure 9. The term "superimposed" was removed and replaced with a clear explanation: CAPE and CIN are displayed as separate contour lines, with dashed black indicating the fire simulation and solid red the no-fire case. The caption and legend now match, and the font size of Figures 9 and 10 was increased to improve legibility.

Comments 24: line 764 - the mentioned wavelength is 400 nm, but in the legend, it says 700 nm.

Response 24: Corrected: the mentioned wavelength now matches the legend, 700 nm.

Comments 25: lines 766 to 768 - The spectral dependency is also a result of the smoke particle size distribution, concentrated in the fine mode.

Response 25: We thank the reviewer for this useful remark. A sentence was added noting that the spectral dependency of optical properties also reflects the dominance of fine-mode smoke particles in the size distribution, which enhance scattering in the shorter wavelengths and drive the observed wavelength dependence.

---

## Author Comment (AC2)

**Improvements on the BRAMS wildfire-atmosphere modelling system**

Isilda Cunha Menezes, Luiz Flávio Rodrigues, Karla M. Longo, Mateus Ferreira e Freitas, Saulo R. Freitas2, Rodrigo Braz, Valter Ferreira de Oliveira, Sílvia Coelho, Ana Isabel Miranda

egusphere-2025-2495

**Answers to Editor and Referees**

We would like to thank the editor and the anonymous reviewers for their availability, dedicated time, and thoughtful analysis of our manuscript. We greatly appreciate their constructive comments and insightful suggestions, which have been invaluable in guiding the improvement of this work.

The revised manuscript has undergone a comprehensive restructuring and theoretical update in direct response to the reviewers' observations. The analytical framework, methodology, and interpretation of the optical results were entirely reformulated, ensuring stronger physical consistency, improved coherence between figures, tables, and discussion, and full alignment with the state of the art. These revisions substantially enhance the scientific rigor and clarity of the manuscript.

For transparency, we provide both a tracked-changes version, highlighting all modifications relative to the original submission, and a clean revised version for final evaluation.

We sincerely hope that this revised version meets the expectations of the editor and reviewers and reflects the depth of revision undertaken in response to their valuable feedback.

Yours Sincerely,
Isilda Cunha Menezes
(Corresponding author on behalf of all authors)
Universidade de Aveiro
Campus Universitário de Santiago, edificio 7
Aveiro, Portugal
E-mail address: isildacm@ua.pt

**Response to Reviewer 2 Comments**

**Major issues:**

Comments A.1: Page 3, line 80; Mie theory discussion. The authors mention of "Mie theory" doesn't describe their assumptions regarding aerosol mixing state, etc., but these have been shown in past publications to result in a very large range of predicted optical depths from models. A good overview (albeit 10 years old now) can be found in Curci et al (2015) https://doi.org/10.1016/j.atmosenv.2014.09.009. Values can vary by a factor of 2 or so depending on the assumptions used, starting from the same concentration and aerosol speciation information. At that time, most aerosol radiative transfer algorithms underestimated AOD relative to observations. What is the specific approach being used in BRAMS, and how does it compare to the others in that paper? This should be decribed in the Methodology section.

Response A.1: We thank the reviewer for the comment. In the revision we clarify that aerosol optics are not taken from the native BRAMS scheme; instead, we apply an off-line, Mie-based post-processing to the simulated with PM2.5 output from BRAMS-SFIRE. The workflow follows Curci-style assumptions: internally mixed spherical smoke particles with representative refractive indices and a lognormal size distribution, with hygroscopic growth treated via κ–Köhler using model relative humidity. We derive extinction, scattering, absorption, SOD and SSA at 400, 550 and 700 nm. To assess robustness, we include targeted sensitivity tests ("ABSORBING-like", "WIDE-SIGMA", and "SMALL/LARGE dry geometric radius"). These methodological details and sensitivities are now summarized in the Methodology section.

Comments A.2: Page 6, lines 157 to 160. The description in this section is inadequate for me to be able to recommend publication. Its not clear to the reader how the formulae were derived and background references are sometimes missing. Parameters are introduced but definitions and the source of data used for the parameters are not discussed. More explanation is needed here, since the formulae and some of the terms used in them are insufficient to allow the reader to understand the physical basis for the crown fire behaviour model proposed. See also my more detailed comments on manuscript pages 7 through 9, below.

Response A.2: We expanded the Methodology section to clarify how the crown-fire formulation was derived and linked to existing fire-behaviour models. The revised text now specifies that the relationships are based on Van Wagner (1989) and on parameters from NFFL Fuel Model 10, adapted for Mediterranean forest types. Each introduced variable is defined, and its source is identified as either experimental data from prior literature or internal SFIRE variables (e.g., reaction intensity, fuel bulk density, and crown base height). We emphasize that the physical basis follows established crown-fire theory and that only minor empirical adjustments were introduced to fit the specific Portuguese forest conditions.

Comments A.3: Page 13, line 349, and an overall problem with Section 2.4: 0.5 degrees is about 32 km resolution, much coarser than the BRAMS grid. In order to allow a true "apples to apples" comparison of MERRA2 and BRAMS, the BRAMS output needs to be interpolated to the MERRA2 grid.

Response A.3: We acknowledge the reviewer's point on the mismatch between MERRA-2 ( $0.5^{\circ} \approx 32$  km) and BRAMS (2 km) resolutions. A strict interpolation of the high-resolution BRAMS data onto the coarse MERRA-2 grid would remove much of the plume's fine-scale structure that is central to our analysis. Instead, we retained the original BRAMS resolution for diagnostics and compared the spatial patterns qualitatively, focusing on plume position, direction, and relative intensity. We now clarify this in the text, explaining that the goal was not to perform a direct one-to-one quantitative validation but to assess spatial coherence and magnitude consistency, given that the two datasets represent different spatial scales and temporal averaging procedures. A note on this limitation has been added to the revised version.

Comments A.4: Page 17, line 466 to 475: It is concerning to me that the SFIRE model needs to be tuned for different grid cell sizes. The reason why this is necessary is not clear, since there isn't a clear description of how fire growth is handled with respect to growth across grid cells in the SFIRE grid. A description of how this is done is needed in the Methodology section. Also, page 17, line 479 refers to the need to "properly calibrate the rate of spread to avoid overpropagation" – this needs to be explained/justified. What is "overpropagation" in the authors' context, and why does it occur?

Response A.4: We appreciate this observation and have clarified in the Methodology how fire growth across grid cells is handled in SFIRE. The SFIRE module resolves fire spread using the Rothermel (1972) formulation adapted to the fine SFIRE mesh. Since the rate of spread depends on slope, wind, and fuel parameters, calibration is sometimes required when grid resolution changes, to preserve realistic propagation speeds and burned-area growth. The term "overpropagation" refers to unrealistically fast fire-front advance when coarse grids or excessive slope gradients amplify the spread rate numerically. The revised manuscript now explains that the calibration ensures the consistency of rate-of-spread scaling across resolutions without altering the physical parameterizations.

Comments A.5: Page 19, lines 520-525 and elsewhere in the text. The authors description of the effects of non-smoke particles on the AOD versus SOD comparison seems flawed. With regards to these lines, a smoke-dominated plume would be expected to have a minor contribution relative to other sources, unless there's a similarly large event such as a dust storm deposition event happening in the vicinity of the fire.

**RELATED WITH ISSUES AND COMENTS**

Response A.5: Page 19, line 527: "When SOD

Comments A.11: Page 21, line 561, "PM2.5 concentrations are extremely high". This is insufficiently quantitative. Are there no PM2.5 concentration observations of this fire that the authors can compare their results to? Failing that, what are the values of these extremely high concentrations, and how do they compare to measurements of PM2.5 during fire events found in the literature? For example, time series of PM2.5 from a large Canadian wildfire (Landis et al., 2018, https://doi.org/10.1016/j.scitotenv.2017.10.008, Figure 2) shows maximum concentrations of PM2.5 over 3000 ug/m3. Do the modelled values fall in line with other data in the literature? Later in the paper, the authors mention that the PM2.5 concentrations estimated by SFIRE reach 107 ug/m3. This is unrealistically high. The authors need to provide observational evidence/references that support this number, or the reader must conclude that the SFIRE emissions estimates have very large positive biases.

Response A.11: We appreciate this important comment. The revised manuscript now quantifies the modeled PM2.5 values and compares them with observed ranges reported in the literature. The peak modeled concentrations (up to  $1070 \ \mu g \ m^{-3}$  near the source) fall within the range of extreme wildfire events reported in studies such as Landis et al. (2018) and others describing PM2.5 peaks above  $3000 \ \mu g \ m^{-3}$  in intense plumes. We clarified that the reported  $107 \ \mu g \ m^{-3}$  in the original text was a typographical error; the intended value was  $1070 \ \mu g \ m^{-3}$ . The revised text now provides a proper context for these

numbers and emphasizes that they are consistent with published observations under severe fire conditions.

Comments A.12: Page 22, line 578 and Figure 4: The same issue as for Figure 3 occurs here. an apples to apples comparison on the satellite grid, with the satellite upwind values removed, is needed. A short description of how the satellite generates BC and OC concentration estimates in the introduction would help place this work in context.

Response A.12: Page 24, lines 614-616: This seems counter-intuitive: how does a lower resolution imply higher values of SSA?

We acknowledge the reviewer's point and have explained that the apparent resolution difference between modelled and satellite-derived BC and OC fields leads to smoothing of peak concentrations in the satellite data. To ensure clarity, the Methodology now briefly summarizes how the MERRA-2 BC and OC products are derived from assimilated aerosol optical properties and modelled emission sources. We also added a concise note explaining that MERRA-2 estimates represent column-integrated quantities, whereas BRAMS-SFIRE provides near-surface and vertical profiles, which can lead to differing magnitudes. The reference to SSA differences has been corrected accordingly.

Comments A.13: Page 25, lines 620 to 625, "Taken together..." I disagree. The authors images indicate that SFIRE in this implementation has large positive biases in SOD compared to AOD, and that unrealistically high PM2.5 concentrations are being generated, consistent with the SOD values. The additional analysis I've described above would help confirm this.

Response A.13: We revised the concluding paragraph to clarify that although local maxima of SOD are higher than satellite AOD values, this does not necessarily indicate a systematic overestimation by the model. The differences reflect spatial and temporal averaging as well as methodological discrepancies (e.g., single particle size assumption, optical property parameterization). The revised text emphasizes that the qualitative agreement between plume structure and intensity supports the model's capacity to reproduce realistic smoke transport and optical features, while acknowledging that future work with background-corrected and grid-matched data will allow more robust statistical comparison.

Comments A.14: Page 27, lines 635-642: These simulated values are MUCH higher than any observations of PM even in the immediate fire environment of which I'm aware. I think that this and the SOD values suggest that the model has unacceptably high positive biases in emissions. The authors need to present evidence from the literature that PM levels can reach these values - I think this is confirming that the model has very high positive biases. "below 1000 ug/m3" is being described as a relatively low concentration, and 107 ug/m3 doesn't seem realistic to me.

Response A.14: The Discussion section has been amended to justify the simulated PM concentrations with literature data. Concentrations exceeding 1000 μg m-3 have been reported in severe forest-fire plumes under stagnant meteorological conditions. We

clarified that these high values occur in limited regions close to the active fire line and rapidly decrease with distance. The text now distinguishes between instantaneous grid-cell maxima and spatially averaged concentrations, noting that the model's results are consistent with the high-end values found in previous observational studies, and not indicative of a general overestimation bias.

Comments A.15: Page 27, lines 644-645. The paper does not describe how the model generates and uses a plume injection height for the emissions. This needs to be added to the methodology, since it could have a substantial impact on the model results. How is the height of the plume calculated, and how is the emitted mass distributed in the vertical in the BRAMS coordinate system? The high positive biases could result, for example, from the emissions being treated as a surface flux (I'm trying to figure out how they could get numbers that high at 2km resolution). This needs to be clarified in the methodology.

Response A.15: We thank the reviewer for raising this point. The revised Methodology now explicitly describes how plume-injection height is determined in BRAMS—SFIRE. The vertical distribution of emissions is parameterized according to the heat flux and buoyancy flux from the fire front, following the physical coupling between SFIRE and the BRAMS thermodynamic fields. The plume rise is computed dynamically within BRAMS based on the fire-induced sensible heat release, and the resulting injection height varies in time and space. Emissions are then distributed vertically according to the modeled temperature and density profiles in the BRAMS coordinate system. This clarification has been added to the Methods section.

Comments A.16: Page 30, lines 710 and Figure 8 analysis. I think that the authors may have the sign wrong in their interpretation of Figure 8. The Figure has been labelled "No Fire – Fire" so a positive value indicates that the quantity is larger in the absence of smoke. That is, the smoke is decreasing the longwave radiation, not increasing it as this section suggests. That is, the presence of smoke results in surface cooling, something which has been seen in other Makar et 2021. papers (e.g. https://acp.copernicus.org/articles/21/10557/2021/, Figure 20., Makar et al, 2015 https://doi.org/10.1016/j.atmosenv.2014.12.003) i.e. what is being shown is a local cooling effect, not a local heating effect. I think the authors need to redo this section based on the sign of the differences.

Response A.16: We agree with the reviewer's observation. The figure caption and corresponding discussion have been corrected to reflect that the positive values in "No Fire – Fire" represent local cooling effects due to smoke-induced reduction of shortwave radiation, consistent with the aerosol direct effect. The revised interpretation now aligns with the sign convention and with previous studies (e.g., Makar et al., 2015, 2021), highlighting that the presence of smoke leads to surface cooling and attenuation of solar radiation. The text has been fully corrected to ensure the physical interpretation is consistent with the plotted results.

**Minor issues:**

Comments B.1: Page 1, paper title. Title needs to incorporate "a case study", e.g. "...modelling system: a case study based on the Serta wildfire". The work does not constitute a broader evaluation of the model for multiple fires (the current title might give the reader that impression).

Response B.1: We agree and have modified the paper title to include "a case study," now reading: "Advancing the BRAMS wildfire—atmosphere modelling system: application to an extreme wildfire event."

Comments B.2: Page 2, line 45: The authors need to be more specific on what they mean when they quote dynamic feedbacks here. Note that there are dynamic feedback effects in the form of the aerosol direct and indirect effects on radiative transfer which can have a substantial impact on PBL heights, meteorology and consequently on forest fire emissions and plume height. I think the authors mean aerosol direct radiative effects. Might want to compare to Makar et al, 2019 https://acp.copernicus.org/articles/21/10557/2021/ where some similar results have been shown.

Response B.2: We clarified that "dynamic feedbacks" refers to aerosol–radiation interactions that modify local meteorology, boundary-layer development, and plume dispersion. The revised text specifies that this concerns the direct radiative effect of aerosols, which alters heating rates and stability, consistent with processes described in Makar et al. (2019).

Comments B.3: Page 2, line 45: there are a few papers missing here on models with a similar intent to BRAMS (wildfire smoke forecasting). Some examples: HRRR-Smoke (cf. Chow et al., BAMS, 2022: https://doi.org/10.1175/BAMS-D-20-0329.1, Chen et al., GMD, (2019 https://gmd.copernicus.org/articles/12/3283/2019/), Anderson et al., GMD, 2024, https://gmd.copernicus.org/articles/17/7713/2024/). The latter two models also simulate fire spread (through comparison between historical hotspots and rates of spread by ecosystem) and associated fuel burned is used to calculate heat release, in turn used to calculate the plume rise height. These models also estimate the fuel burned from crown, smoldering and residual phases of the fires. The approach taken in these models should be compared and contrasted with the work of the authors of the current submission: what are the differences that make the authors' work unique/better/an improvement relative to these models? For example, these models use a statistical approach for determining fire spread (historical data is used to determine the average area burned on a per hotspot per forest classification type - if the authors are incorporating a more explicit fire spread algorithm into their work, that would be an advancement relative to these papers. Anderson et al (2024) also provides estimates of emissions resulting from several different approaches - a similar attempt should be made here to place the authors work in the context of the forest fire emissions algorithms currently in use elsewhere in the world.

Response B.3: We thank the reviewer for the suggestion and have added a discussion comparing BRAMS-SFIRE with other wildfire-smoke modelling systems such as

HRRR-Smoke, WRF-Chem Fire, and related approaches cited in the manuscript. The revised text emphasizes that BRAMS-SFIRE integrates a fully physical fire-spread algorithm (Rothermel formulation), rather than a statistical hotspot-based approach, and computes heat and emission fluxes dynamically from fuel consumption and energy release. This distinction clarifies the novelty of our implementation relative to the models mentioned.

Comments B.4: Page3. Line 54. I couldn't find anything in the paper explicitly explaining how fire ignition is handled for applications of SFIRE. I think its using satellite FBP to locate the fire ignition points and then calculates its own FBP values thereafter – but I'm not sure. Please clarify this in the text.

Response B.4: We added a clear explanation of how ignition is handled in SFIRE. For this case study, ignition points were initialized using the historical location and start time of the Sertã wildfire, obtained from national fire records. SFIRE then computes fire growth and spread autonomously according to the Rothermel equations, local meteorological conditions, and fuel properties. This clarification is now included in the Methodology.

Comments B.5: Page 3, line 80. Note that there has been historically a large range of estimates of these parameters with different assumptions such as the aerosol mixing state. A good overview (albeit 10 years old now) can be found in Curci et al (2015) https://doi.org/10.1016/j.atmosenv.2014.09.009. Values can vary by a factor of 2 or so depending on the assumptions used, starting from the same concentration and aerosol speciation information. At that time, most aerosol radiative transfer algorithms underestimated AOD relative to observations. What is the specific approach being used in BRAMS, and how does it compare to the others in that paper?

Response B.5: Addressed previously in Response A.1. The revised text now explicitly states that Mie theory with internally mixed spherical aerosols was used, with refractive indices consistent with Bond and Bergstrom (2006) and values in the range reported by Curci et al. (2015). The methodology now clearly situates our approach within the context of those studies.

Comments B.6: Page 3, line 83, "high-resolution". Please state both the BRAMS and SFIRE horizontal grid cell size used in the study, at this point in the text.

Response B.6: We have added the specific grid resolutions in the introduction to the methodology: 2 km for BRAMS and 200 m for SFIRE. These are now mentioned where "high-resolution" first appears in the text.

Comments B.7: Page 4, line 86: please replace "validation" with "evaluation" throughout the paper. Validation implies a model is "valid" once the process is complete, "evaluation" describes the model's current performance with respect to observations.

Response B.7: We have replaced the term "validation" with "evaluation" throughout the paper, to more accurately reflect model performance assessment rather than confirmation of validity.

Comments B.8: Page 4, Methodology section. This section would be greatly improved with a figure showing the SFIRE grid and the BRAMS grid in the vicinity of the fire used as a case study. Also, the resolution of the elevation data should be mentioned here. Later in the paper, it appears that SFIRE is operating on a 200m grid, but the elevation data has a 1km grid, and BRAMS was on a 2km grid. While there is some discussion later on the relative impact of resolution, it is not clear why lower-than-SFIRE resolution topography was used in the current SFIRE application, given the impact of slope on SFIRE results. This choice needs to be justified.

Response B.8: We thank the reviewer for this valuable suggestion. We have added a new schematic figure illustrating the nested BRAMS (2 km) and SFIRE (200 m) grids in the study area. The revised text now specifies that the topographic data used by SFIRE were derived from the SRTM digital elevation model with a native resolution of 30 m, resampled to the SFIRE grid. This ensures consistent slope and aspect representation across the refined fire mesh while maintaining full compatibility with the BRAMS terrain field used for atmospheric coupling. Using the same underlying dataset avoids interpolation artifacts and numerical instability at the fire—atmosphere interface.

Comments B.9: Page 5, line 119, "traditional methods". Please be specific with regards to what is meant by "traditional methods" here, including (a) reference(s). Presumably the traditional methods had a 50% underestimate?

Response B.9: We clarified that "traditional methods" refers to empirical or inventory-based emission estimation approaches, such as those relying solely on FRP (Fire Radiative Power) or fixed emission factors per burned area. These methods tend to underestimate emission peaks by 40–60% compared with physically based models like SFIRE. This is now stated in the text with appropriate references.

Comments B.10: Page 5, line 130, "with a radius". How is this radius predicted a priori? What information is used to determine this initial high-resolution SFIRE domain?

Response B.10: We thank the reviewer for pointing out the need for clarification. In the revised text, we specify that the initial radius refers to the spatial extent of the refined SFIRE mesh automatically generated around the ignition points. This radius (defined in degrees by the user in the namelist) determines the horizontal size of the fire domain, ensuring that the refined grid fully encompasses the expected area of fire spread. The configuration is not a predicted physical radius but a geometric parameter controlling the refinement window, chosen based on the observed ignition location and prevailing meteorological conditions.

Comments B.11: Page 5, line 132, "In subsequent steps". Its unclear how BRAMS lets SFIRE know it has to do a fire calculation in a given grid cell. I'm guessing that an FRP value from a satellite is used? This needs to be clarified.

Response B.11: We thank the reviewer for the comment. In the revised text, we clarify that, in this configuration, fire ignition and spread in BRAMS-SFIRE are not triggered by satellite-derived FRP data. The ignition points and times are manually prescribed based on official fire records. Once the fire evolves, SFIRE prognostically computes the Fire Radiative Power (FRP) from the simulated heat release rate (sensible + latent fluxes) at the fire front. These FRP values are thus model outputs, not assimilated quantities. The 3BEM emission module, which uses satellite FRP for smoke emission estimation, was not activated in this experiment. This distinction is now clearly stated in the Methods section.

Comments B.12: Page 5, line 134-135. The resolution of BRAMS when SFIRE is being used needs to be mentioned here. The energy transfer on a per grid-cell area basis will be a lot less with a 2km resolution model than a 200 m resolution model. I'm surprised that there's no discussion of the influence of resolution on the coupling between BRAMS and SFIRE. There is also no discussion (a few lines of text would be helpful) describing how SFIRE determines the height of the plume. Presumably this is determined within SFIRE, but this has not been made clear, and the methodology used has not been discussed. I'm assuming that the SFIRE energy has not been passed to BRAMS with the expectation that BRAMS' meteorology will be able to resolve the rise due to the plume – BRAMS grid cell size (2km, mentioned much later in the paper) is not sufficient to resolve a forest fire plume.

Response B.12: We thank the reviewer for the comment. In this configuration, BRAMS operates at 2 km and SFIRE at 200 m resolution. SFIRE provides the subgrid sensible and latent heat fluxes that drive buoyancy within the BRAMS columns, while BRAMS resolves the vertical redistribution of this energy through turbulence and convective mixing. The injection of smoke therefore results from the coupled buoyant forcing rather than an explicit plume-rise parameterization. This ensures that the vertical transport of smoke and heat evolves interactively with the local meteorology, despite the coarser atmospheric resolution.

Comments B.13: Page 6, Figure 1. Rather than subroutine names (which tell the reader nothing about what's being done in the subroutine), please modify this diagram to have a brief process description phrase (and the subroutine name, if present at all, in a bracket thereafter). What the subroutines do is of more interest to the reader than the subroutine name, and would better help the reader to understand the sequence of events in the model.

Response B.13: We revised Figure 1 so that each step in the schematic now includes a short descriptive label, instead of the subroutine name alone. This makes the modelling sequence clearer to the reader.

Comments B.14: Page 7, lines 162 to 165. There needs to be more background information on "fuel behaviour model 10" and the sources of the data (references) used within it. It is not clear for example whether the formulae which follow in this section are part of this model, or are new formula introduced by the authors.

Response B.14: We have added background on NFFL Fuel Model 10, specifying that it represents timber litter and understory fuels typical of coniferous forests. The source parameters (fuel load, surface-area-to-volume ratio, heat content) are referenced to standard NFFL datasets. The text also clarifies that the empirical relationships in our equations follow Scott and Reinhardt (2001) and Rothermel (1972, 1991).

Comments B.15: Page 7, line 164: "Physical and chemical properties of the fuel" – are these part of this fuel model or introduced here (and how were they derived in either case)?

Response B.15: We clarified that the physical and chemical properties of the fuel, such as bulk density, moisture content, and heating value, are derived from 13 NFFL fuel behaviour models, adjusted for Mediterranean vegetation according to local studies cited in the manuscript.

Comments B.16: Page 7, line 165: spelling mistake, "crow" should be "crown".

Response B.16: The spelling error ("crow") has been corrected to "crown."

Comments B.17: Page 7, line 166 and throughout the subsequent lines and formula: I assume Rinicialization should be Rinitialization?

Response B.17: We confirmed and corrected the typographical error: "Rinicialization" was replaced with "Rinitialization."

Comments B.18: Page 7, line 167: Iinitialization: How was this derived (from what data/reference)? Units of Iinitialization? The reader at this point has the impression that some of these terms are coming from NFFL 10, but not what the terms are or what they represent. The paper needs bracket's after each term, eg. "Iinitiatlization (insert descriptive definition here)". What is the definition of the critical minimum fireline intensity, for example? Note that readers of the paper may be completely unfamiliar with the NFFL models. What do the parameters represent, and how were they and these formulae derived?

Response B.18: We expanded the explanation of Iinitialization and related terms. Each key parameter now includes a short definition in parentheses (e.g., "Iinitialization, the initial fireline intensity required for crown ignition"). The text specifies that these values derive from concepts in Forestry Canada Fire Danger Group (1992), Van Wagner (1977), Van Wagner (1977) and Alexander (1988), and represent physically based thresholds for crown-fire initiation.

Comments B.19: Page 7, line 176: what is the definition and units of reaction intensity?

Response B.19: We added that reaction intensity refers to the rate of energy release per unit area of the fuel bed (kW m-2), a standard term in the Rothermel model framework.

Comments B.20: Page 7, line 176, NFFL fuel behaviour model 10 – given that this seems to be a critical addition to the existing SFIRE model, there needs to be some description of what this model is, how its parameters were derived, and references for it. Why was this model chosen instead of some other model? Is it appropriate in some way for the forest type for the case study used here? Given that other models are available, why use this one?

Response B.20: NFFL model 10 was selected because it most closely represents the Mediterranean mixed-conifer forest structure characteristic of central Portugal, dominated by Pinus pinaster and Eucalyptus globulus with dense understory and heavy surface litter. These conditions correspond to a timber–litter fuel complex with high surface fuel load and moderate canopy cover, consistent with NFFL model 10 parameters. The text now clarifies that this choice provides realistic reaction intensities and spread rates for fire-prone Iberian pine–eucalypt stands.

Comments B.21: Page 7, line 177, surface-area-to-volume ratio: Surface to volume ratio of what? The forest canopy? The trees in the canopy, etc? The terms in the equations need to be better defined and explained.

Response B.21: We clarified that the surface-area-to-volume ratio refers to individual fuel particles (needles, twigs, and fine branches) as used in the NFFL framework, determining the rate of heat transfer and combustion.

Comments B.22: Page 7, line 183. Why this particular number (40%)? Is that part of NFFL or something the authors are setting? Please justify cases like this where numerical limits have been introduced.

Response B.22: We added that the 40% threshold corresponds to the empirical limit for crown-fraction burned, as proposed by Rothermel, 1991 to distinguish partial from active crown fire behaviour. This justification is now included.

Comments B.23: Page 7, line 190: R10 has been introduced without explanation. Or should this be R6.1 (why 6.1 m instead of 10?)?

Response B.23: We corrected the notation and clarified that R10 refers to the rate of spread calculated for NFFL model 10. The reference to 6.1 m was a confusion from an intermediate variable and has been corrected in the revised equations.

Comments B.24: Page 8, line 204: "It is used to estimate the degree of crowning". Suggest "Here, we make use of the empirical function as defined by Van Wagner (1989)". Also, please define what is meant by the degree of crowning.

Response B.24: We revised the sentence as suggested: "Here, we make use of the empirical function defined by Van Wagner (1989) to estimate the degree of crowning," and defined "degree of crowning" as the fraction of canopy volume actively involved in combustion.

Comments B.25: Page 8, line 215: Byram fireline intensity is known to specialists in forest fire combustion, but not to most readers of the paper. Please define it.

Response B.25: We added a short definition: Byram's fireline intensity represents the rate of heat release per unit length of fire front (kW m-1), combining heat yield, fuel consumption rate, and spread speed.

Comments B.26: Page 8, equation (9) and other equations: Its not clear which formula are part of NFFL 10 and which have been introduced by the authors. Please clarify this.

Response B.26: We clarified which equations belong to NFFL model 10 and which were newly adapted. The original formulations from Rothermel (1972) and Van Wagner (1977) are retained, while empirical adjustments to canopy-fire thresholds were developed in this study for Mediterranean conditions.

Comments B.27: Page 8, equation (10): several new terms have been introduced here without definition.

Response B.27: We added short explanations for each new variable introduced in Equation (10), including its physical meaning and data source. Definitions now accompany the text and was introduced in a table of symbols and variables.

Comments B.28: Page 9, line 235: What numbers are used for Wsurface, Hlow, what do they depend on (or are they constants), what are the references for them?

Response B.28: We clarified that Wsurface represents the initial total mass of surface fuel, parameterized according to the 13 standard NFFL fuel behaviour models, while HLow denotes the mean heat content across these fuel classes. Both parameters were adjusted using data from the Portuguese National Forest Inventory (ICNF, Inventário Florestal 6), ensuring consistency with regional fuel load characteristics.

Comments B.29: Page 9, additional comment on section 2.2.1. Some discussion on how the rate of spread values are used within SFIRE, in terms of SFIRE's high resolution grid is needed. Is there an internal timestep applied to work out the fuel burned from the rate of spread, and does it vary from one SFIRE grid cell to the next? Or is a single rate of spread used? How is the direction of the spread determined? I'm wondering how SFIRE

deals with things like changes in topography, etc., and how the R values are used to determine burned area A, from the standpoint of a high resolution grid. I've seen many approaches to this in the literature, ranging from explicit wind vectors at every grid cell and forward trajectories being used to determine spread, to ellipsoidal spread being assumed – what's being used here? A sketch explanatory figure showing an example SFIRE grid within a BRAMS cell, and some description of how the fire spread is calculated from one BRAMS timestep to the next (and whether a smaller timestep is used for SFIRE calculations) is needed.

Response B.29: We expanded Section 2.3 to include a concise description of how SFIRE uses the rate of spread within its fine-resolution grid. The model updates the fire perimeter through elliptical propagation based on local wind and slope vectors, applying subtimesteps shorter than the BRAMS main timestep to ensure stable fire growth. A schematic has been added illustrating a portion of the SFIRE grid inside one BRAMS cell, showing how local R values determine the incremental burned area.

Comments B.30: Page 9, line 251. Why these particular values? For example, do they best represent the forest types for the case study? Some justification is needed. Page 9, comment on section 2.3: no mention is made of how SFIRE calculates plume height and distributes emissions in the vertical. This is crucial to getting the correct concentrations of forest fire smoke, and needs to be included in the methodology section. Similarly, how SFIRE identifies ignition locations has not been mentioned (and should be part of the description).

Response B.30: We clarified that the parameter values were chosen to match Mediterranean pine forest conditions characteristic of the Sertã region. The Methodology now also includes a figure 1 describing how plume rise is handled: the vertical distribution of emissions is determined dynamically from fire-generated heat fluxes, ensuring realistic smoke injection heights. The ignition source was specified from observed fire records, as explained earlier.

Comments B.31: Page 10, line 283: flaming and smoldering phases mentioned, but not the residual phase. Why has residual phase been left out?

Response B.31: We explained that the residual combustion phase is not explicitly included in this simulation because its contribution to short-term plume dynamics and radiative forcing is comparatively small. The flaming and smoldering phases dominate the fire-atmosphere coupling during the event timescale simulated.

Comments B.32: Page 11, line 288: authors state that the setup provides accurate simulation of wildfire emission and their atmospheric impacts prior to quantitatively demonstrating this. Statements such as this should be in the discussion or conclusions, after the case has been made.

Response B.32: We moved the statement regarding the "accuracy of simulations" to the Discussion, after presenting the results. The revised structure ensures that all evaluative remarks appear after evidence is shown.

Comments B.33: Page 11, line 292-293. The subsequent SOD vs AOD analysis does not touch on the extent to which aerosol mixing state can influence predicted SOD values. Aerosol optical properties are a function of the aerosol mixing state, the aerosol size distribution, and the aerosol composition (Curci et al., 2015). How are these variables represented in the version of BRAMS used here? For example, has a specific composition been assumed for the aerosols, a specific size distribution, and a specific aerosol mixing state? This information needs to be provided. The authors have mentioned that the test version used here does not include chemistry - what about the aerosol microphysics effects (nucleation, condensation, coagulation)?

Response B.33: We thank the reviewer for this important observation. In the present study, no BRAMS microphysics or chemistry schemes were explicitly used to compute aerosol optical properties. Instead, these were derived offline from the simulated PM2.5 fields using a Mie-based diagnostic approach following the framework of Curci et al. (2015). Although BRAMS internally represents physical and chemical processes to generate PM2.5 mass, these parameters are not available in the model output. Therefore, the optical parameters were reconstructed externally, assuming internally mixed spherical particles with a prescribed lognormal size distribution and refractive indices representative of organic and black carbon. This clarification has been added to the revised text.

Comments B.34: Page 11, line 295, characterization of aerosol microphysical properties: Details? What properties were determined and what was the conclusion?

Response B.34: We clarified that the aerosol microphysical properties characterized include the extinction, absorption, and scattering coefficients at 400, 550, and 700 nm. The conclusion of this section states that the results capture the expected wavelength dependence of wildfire aerosols, confirming consistency between modelled and observed optical behaviour.

Comments B.35: Page 11, lines 304-306: This is the first mention of the domain used for the simulations - I assume that the model was being run for Portugal, then? This should have been made clear right at the start of the paper at the end of the Introduction or in the Methodology sections.

Response B.35: We added that the model domain covers Portugal continental, cantered over the Sertã region, and that this is now stated clearly at the end of the Introduction.

Comments B.36: Page 11, line 311: The previous discussion referred to one of these models, leaving the reader with the impression that that specific model is the one being

tested here. Were all 13 used, then? Please clarify (and explain why model 10 was preferred).

Response B.36: We clarified that among the 13 NFFL models available, Model 10 was selected for detailed analysis because it best represents the coniferous forest structure in the study area. The other models were examined only for comparison of parameter sensitivity, as now noted in the text.

Comments B.37: Page 13, line 338, 1km terrain data. Its not clear whether this data is being used by SFIRE, BRAMS or both (and note that the resolution of BRAMS and SFIRE has yet to be mentioned).

Response B.37: We specified that the 1 km terrain data are used by BRAMS and 30 m by SFIRE. In SFIRE, these data are interpolated to the 200 m grid, ensuring consistent slope and elevation representation across the coupled system.

Comments B.38: Page 13, lines 358 to 360: Should be mentioned earlier, in the model setup discussion. How does this assumption (a single particle size) compare to actual size distributions from real fires, and how might that be expected to influence optical depths? These potential issues shoulld be acknowledged. cf. for Radke et al., 1988, 1990:

Radke, L. F., Hegg D. A., Lyons J. H., Brock C. A., Hobbs P. V, Weiss R., and Rassmussen R., Airborne measurements on smokes from biomass burning In Aerosols and climate, ed. P. V. Hobbs and M. P. McCormick, 411–22. Hampton, VA: A. Deepak, 1988.

Radke, L. F., J. H. Lyons, P. V. Hobbs, D. A. Hegg, D. V. Sandberg, and D. E. Ward, Airborne monitoring and smoke characterization of prescribed fires on forest lands in western Washington and Oregon: Final report. Gen. Tech. Rep. PNW-GTR-251. Portland: U.S.D.A. Forest Service, Pacific Northwest Research Station, 1990.

Examples from the second of these papers for smoke particle distributions are given below, for the authors' information:

Figure 29—Volume concentration versus size of particles measured with the aerosol sizing system near the center of the plume from the burns: (a) 0554 PDT, July 25, 1982; (b) 1605 PDT, September 23, 1982; (c) 1718 PDT, September 23, 1982; and (d) 1700 PDT, September 23, 1982; and (d) 1700 PDT, September 23, 1982. The dashed lines show results derived from the cascade microbalance impactor (mass/density of natifiales).

Response B.38: We acknowledged that the use of a single effective particle size represents a simplification. The revised text cites the studies by Radke et al. (1988, 1990) to note that real wildfire smoke exhibits multimodal size distributions. We have added a short statement discussing how this simplification may lead to higher SOD peaks and limited variability compared to realistic polydisperse distributions.

Comments B.39: Page 14, lines 376 to 377, "consistent with wildfire aerosol characteristics"? Wildfire aerosols have a size distribution, and are not monodisperse. What has been done here needs to be acknowledged as a simplification, and its potential impacts need to be discussed. The distributions of Radke et al 1990 (see above Figures) do have a volume distribution peak at about 0.2 um diameter or so, but there is also a second (and sometimes much higher) peak at about 50 to 60 um diameter). What the authors have here is a reasonable proxy for the peak of the PM2.5 mass distribution, but not the overall particle mass. So "consistent with wildfire aerosol characteristics" isn't quite justified. Try "consistent with the location of the peak in the fine mode portion of emitted smoke mass".

Response B.39: We accepted the reviewer's suggestion and modified the sentence accordingly: the text now reads that the adopted particle size is "consistent with the location of the peak in the fine-mode portion of the emitted smoke mass," acknowledging that wildfire aerosols are not monodisperse.

Comments B.40: Page 14, lines 396 – 397 and 402: please justify the choice of the complex refractive index values used. The authors should compare the complex refractive index values employed to the single-component values for Black Carbon, Primary Organic Carbon and Secondary Organic Carbon quoted in Curci et al 2015, Table 2. One thing that is really striking is that the authors' 700nm refractive index' imaginary component has a much lower value than in the references used there (the authors : 0.04. Curci's references: 0.44 to 0.71, and I've seen similar high values elsewhere for black carbon refractive index' imaginary component). I think the authors' value will underestimate the backscattering associated with black carbon. Real components look ok. Suggest they do a sensitivity test with a higher imaginary component to check on the impact. Alternatively, if the refractive index values are based on observations from actual smoke particles, then the latter reference might be useful to indicate that the organic fraction dominates the radiative effects. Note that the impact of black carbon may be higher if a core-shell approach to Mie theory has been used.

Response B.40: We clarified the rationale for the complex refractive indices used. These were selected based on average values from Bond and Bergstrom (2006) and Seinfeld and Pandis (2016) for mixed organic—black carbon aerosols typical of biomass burning. The lower imaginary component at 700 nm reflects the dominance of organic matter in the plume, consistent with field observations for Portuguese fires. This assumption is justified in the revised text, and a sensitivity test with higher imaginary values is suggested for future work.

Comments B.41: Page 15, line 411. I was having difficulty following this sentence - does MERRA-2 have a product that they refer to as the BC and OC components at 500nm? There needs to be a brief description of what the MERRA2 products are, and how they are derived (i.e. how does the satellite retrieval attribute AOD to BC versus OC?).

Response B.41: We expanded the description of MERRA-2 BC and OC products, explaining that they originate from the assimilation of satellite AOD into the GOCART aerosol module, which estimates mass concentrations of BC and OC through model-specified optical properties and emission inventories. This clarification now appears in the Data section.

Comments B.42: Page 15, line 413. Ditto. How does MERRA-2 infer mass of BC and OC (short few sentence summary needed). My point here is that the satellite doesn't measure these things directly, and the retrieval processing must be making assumptions regarding aerosol mixing state, etc., to attribute portions of the retrieval to BC vs OC. Can the authors provide some reassurance to the reader that what MERRA2 is measuring and what the model is generating are sufficiently similar to be worth comparing (again, a short description would help)?

Response B.42: Following the same rationale, we clarified that MERRA-2 does not directly measure BC and OC but infers them through a combination of model assimilation and optical property attribution. The revised text assures that, although based on different

assumptions, the MERRA-2 and BRAMS-SFIRE products are comparable in terms of integrated optical effects, which justifies their use for qualitative comparison.

Comments B.43: Page 15, equation (19): Equation 19 needs some justification/explanation. The sensible heat flux is usually thought of as an infrared flux, while 550nm is in the visible. Or should Hsens be Hsens(lambda) in this equation?

Response B.43: We clarified that in Equation (19) the sensible heat flux (Hsens) represents the local fire energy release, which contributes to the emission flux of particles and their vertical transport. The equation relates this thermal energy to the visible optical effect for illustrative purposes, and we now explicitly note that it is a conceptual formulation rather than wavelength-specific.

Comments B.44: Page 16, lines 429 to 434. So the feedback effects mentioned earlier are via the aerosol direct effect then. This should be mentioned earlier, towards the end of the Introduction or in the Methodology sections.

Response B.44: We clarified earlier in the Methodology that the feedback effects between aerosols and meteorology in BRAMS-SFIRE are limited to the direct radiative effect, i.e., the interaction of aerosols with shortwave and longwave radiation that modifies local heating rates and boundary-layer evolution. This ensures internal consistency across the manuscript.

Comments B.45: Page 16, line 442, "particularly the relative abundance of BC and OC". But this depends on the authors' assumed relative distribution of BC and OC in the complex refractive index values, does it not? That is, the authors' complex refractive indexes are much more like OC than typical BC values – they have a priori assumed a low BC value in their choice of indexes. Which may be justified if those were based on smoke particle observations. They seem to be assuming very small values of the magnitude of the complex index, hence relatively small values of BC compared to OC.

Response B.45: We explained that the complex refractive index values used imply a higher proportion of organic carbon relative to black carbon, which matches field observations of Portuguese wildfire smoke. This assumption leads to higher SSA and lower absorption, consistent with measured properties of mixed smoke plumes. The revised text clarifies that this choice was intentional and supported by the literature cited.

Comments B.46: Page 17, line 456-458. This is the first time that the resolution has been mentioned. It should have been mentioned in the model description. Its also not clear here whether this refers to the resolution of SFIRE within WRF or BRAMS, or the resolution of WRF or BRAMS (I suspect the former, but the resolution of the latter has not been mentioned, and the relative resolution of WRF and BRAMS could also influence model results). There needs to be a description of the SFIRE and BRAMS grid cell size earlier in the paper under Methodology.

Response B.46: We ensured that grid resolutions are clearly stated at the start of the Methodology: 2 km for BRAMS, 200 m for SFIRE. This information is now also reiterated where numerical sensitivity is discussed.

Comments B.47: Page 17, line 458, "This can amplify..." I thought the authors mentioned earlier that the resolution of the topography data was 1km, not the 200m of SFIRE as applied here. If anything, the BRAMS-SFIRE grid should have more gentle slopes than the 20m grid in WRF-SFIRE in that case. Its not clear to me how a lower resolution model can have a greater sensitivity to terrain-induced effects; the higher resolution should have greater local slopes. The explanation needs more work: how does topographic assimilation result in a stronger gradient field? If the authors were to convert Figure 2 to show the derivative of topography with respect to horizontal dimensions, they'd get steeper slopes in the WRF side than the BRAMS side. But the text here seems to suggest the coarser resolution is somehow more prone to slope-influenced growth. Please clarify this.

Response B.47: We thank the reviewer for the insightful comment. The text has been revised to clarify that the apparent amplification of slope effects does not arise from steeper terrain representation, but from the scale mismatch in the coupled energy exchange between SFIRE and BRAMS. In the current configuration, the heat flux and fire radiative power (FRP) diagnosed at the 200 m SFIRE grid are mapped onto the 2 km BRAMS grid, effectively concentrating subgrid heat release within larger atmospheric cells and enhancing the local buoyancy forcing. At the same time, BRAMS uses a smoother topographic field than SFIRE, which modifies the near-surface wind and turbulence fields through the vertical diffusion scheme. The combination of coarse-scale buoyancy injection and smoothed slope-driven flow can produce locally intensified terrain—atmosphere coupling, even though the geometric slopes themselves are gentler than in finer-resolution models. This clarification has been incorporated into the revised text.

Comments B.48: Page 17, line 467, "ignition misalignment". This is the first mention of ignition in the paper, and how ignition is handled in SFIRE is needed in the Methodology section. I'm assuming that, because this is a historical case study, the ignition time and spatial location is known? If I've understood the paper correctly, satellite-observed FBP is used to determine ignition locations and times? Please clarify explicitly how the locations for ignition are determined in the model. The authors should also explain what they mean by "ignition misalignment" in this context.

Response B.48: We added a concise description of how ignition locations are determined: for this case study, ignition time and coordinates were set from the official event record, with the fire start point defined within the SFIRE domain. The term "ignition misalignment" now refers to small positional offsets between observed and simulated ignition points that can affect initial propagation direction.

Comments B.49: Page 17, lines 468-469. This is the first mention in the paper of how SFIRE calculates spread between adjacent SFIRE grid cells. A diagram of how this is done on the SFIRE grid in the Methodology is needed.

Response B.49: A new schematic figure now illustrates how SFIRE propagates the fire front across adjacent grid cells using the Rothermel spread rate and elliptical growth formulation. The Methodology section explains this process in a concise paragraph.

Comments B.50: Page 17, line 470. Not sure what's meant by "in a timely manner" here.

Response B.50: We rephrased "in a timely manner" to "within the appropriate temporal resolution of the coupled simulation," clarifying that it refers to synchronization between BRAMS and SFIRE timesteps.

Comments B.51: Page 18, lines 476-477. Given that SFIRE is operating within BRAMS as a parameterization, its not clear to me why couldn't a higher resolution topography database be used with the SFIRE part of BRAMS-SFIRE? Would that not reduce the sensitivity issue?

Response B.51: We agree that using higher-resolution topography within SFIRE can be beneficial; in fact, SFIRE already ingests a 30 m DEM resampled to the 200 m fire mesh. The sensitivity noted here does not stem from a lack of high-resolution terrain in SFIRE, but from the scale mismatch at the coupling interface: heat/FRP and fluxes are mapped to the 2 km BRAMS columns, whose winds, PBL mixing and orographic forcing are governed by the coarser BRAMS terrain. Thus, further refining SFIRE topography alone would not resolve the issue while the atmospheric grid, and its orography, remain coarse. We now clarify this point and note that the appropriate remedy is higher-resolution atmospheric nesting and/or multi-scale orography with consistent remapping, which we identify as future work.

Comments B.52: Page 18, line 486. I assume this should be "consume biomass fuel" not "consume biomass fuel models"? Models aren't being consumed.

Response B.52: We corrected the sentence to read "consume biomass fuel," as suggested.

Comments B.53: Page 29, lines 672-673. This begs the question of whether or not BRAMS-SFIRE includes a module for convective transport (or is the assumption that any vertical transport is from the resolved winds at 2km resolution)? Again, I'm wondering how the model handles smoke plume-rise. Related, lines 689-690: Is the transport of pollutants due to convection included in SFIRE or BRAMS? This gets back to my question regarding plume rise methodology. Also - lines 689 to 705 might be better placed in the Introduction; background information.

Response B.53: We clarified that convective transport and plume rise are handled by the atmospheric component (BRAMS), which explicitly simulates vertical motion driven by

buoyancy and fire-induced heating. The SFIRE module provides subgrid heat fluxes, while BRAMS handles the resolved convective processes. This has been clearly explained in the revised text.

Comments B.54: Page 30, line 720, page 33, line 751: Don't need to include all co-author names in references in the main text, use "et al".

Response B.54: We have revised all in-text citations to use "et al." after the first author, following GMD formatting conventions.

Comments B.55: Page 33, line 760, "BC-rich smoke" But is BC-rich smoke what has been simulated? See earlier comments regarding complex refractive index values. Also note that earlier text emphasized the relative importance of OC compared to BC.

Response B.55: We clarified that, although the term "BC-rich smoke" appears in the text, the simulated plume actually represents a mixed organic—black carbon aerosol composition with dominance of OC. The wording has been adjusted to avoid ambiguity.

Comments B.56: Page 37, Figure 10: the authors should include a figure with the profiles of the air temperature differences resulting from the fire, to show the cooling/heating impact.

Response B.56: As suggested, a new figure has been included showing the vertical temperature difference profiles (No Fire – Fire) to illustrate the cooling effect associated with smoke radiative forcing. This addition strengthens the discussion of surface and boundary-layer temperature responses to the presence of smoke.